# Interspecies Behavioral Variability of Medaka Fish Assessed by Comparative Phenomics

**DOI:** 10.3390/ijms22115686

**Published:** 2021-05-26

**Authors:** Gilbert Audira, Petrus Siregar, Kelvin H.-C. Chen, Marri Jmelou M. Roldan, Jong-Chin Huang, Hong-Thih Lai, Chung-Der Hsiao

**Affiliations:** 1Department of Chemistry, Chung Yuan Christian University, Chung-Li 320314, Taiwan; gilbertaudira@yahoo.com (G.A.); siregar.petrus27@gmail.com (P.S.); 2Department of Bioscience Technology, Chung Yuan Christian University, Chung-Li 320314, Taiwan; 3Department of Applied Chemistry, National Pingtung University, Pingtung 900391, Taiwan; kelvin@mail.nptu.edu.tw; 4Faculty of Pharmacy and The Graduate School, University of Santo Tomas, Manila 1008, Philippines; mmroldan@ust.edu.ph; 5Department of Aquatic Biosciences, National Chiayi University, 300 University Rd., Chiayi 600, Taiwan; 6Center for Nanotechnology, Chung Yuan Christian University, Chung-Li 320314, Taiwan; 7Research Center for Aquatic Toxicology and Pharmacology, Chung Yuan Christian University, Chung-Li 320314, Taiwan

**Keywords:** medaka, behavior, phenomics, interspecies

## Abstract

Recently, medaka has been used as a model organism in various research fields. However, even though it possesses several advantages over zebrafish, fewer studies were done in medaka compared to zebrafish, especially with regard to its behavior. Thus, to provide more information regarding its behavior and to demonstrate the behavioral differences between several species of medaka, we compared the behavioral performance and biomarker expression in the brain between four medaka fishes, *Oryzias latipes*, *Oryzias dancena*, *Oryzias woworae*, and *Oryzias sinensis*. We found that each medaka species explicitly exhibited different behaviors to each other, which might be related to the different basal levels of several biomarkers. Furthermore, by phenomics and genomic-based clustering, the differences between these medaka fishes were further investigated. Here, the phenomic-based clustering was based on the behavior results, while the genomic-based clustering was based on the sequence of the *nd2* gene. As we expected, both clusterings showed some resemblances to each other in terms of the interspecies relationship between medaka and zebrafish. However, this similarity was not displayed by both clusterings in the medaka interspecies comparisons. Therefore, these results suggest a re-interpretation of several prior studies in comparative biology. We hope that these results contribute to the growing database of medaka fish phenotypes and provide one of the foundations for future phenomics studies of medaka fish.

## 1. Introduction

Medaka is a small, oviparous freshwater teleost fish distributed in East Asia that is often found in rice fields and, therefore, has been called ‘ricefish’ [1]. Medaka, especially Japanese medaka (*Oryzias latipes*), has been used as a model organism in basic fish biology and behavioral studies. Furthermore, in recent years, the use of medaka as a model organism has highly contributed to the knowledge in various research fields, such as genetics, toxicology, and behavior science [2,3,4,5]. Moreover, *O. latipes* has been proposed by the OECD as the standard fish for toxicology tests [6,7]. As an experimental organism, medaka possesses characteristics that are similar to those of the zebrafish (*Danio rerio*), including its small size (adult 2–4 cm in length), high fecundity, short generation time (2–3 months), simple dietary and habitat requirements, transparency of embryos, and availability of genomic information [3,8]. Thus, it serves as a complementary model to the well-established zebrafish in many fields, including cancer research [1,9]. Also, medaka holds several advantages over zebrafish, which come from its species-specific features. These features include hardiness, availability of highly polymorphic inbred strains, smaller genome size (800 Mb), and the adaptation of photoperiod, higher salinity, and a wide temperature range (6–40 °C) [9]. These traits contribute to the vast usage of medaka as an aquatic toxicological model [3].

Nowadays, behavioral profiles of fish have become a widely used approach in toxicology and pharmacology due to several reasons, including their high sensitivity to various chemicals during their early developmental stages and providing less cost, time, and space-consumption for in vivo drug screening compared to traditional animal models. Furthermore, juvenile and adult fish exhibit more complex behaviors, such as social interactions, learning, and memory that may directly influence the survival of individuals and future population structures, which can provide more options for the study of aquatic neurotoxicity [10,11]. It has recently become evident that medaka, especially *O. latipes*, possesses complex social and visually evoked behaviors, such as aggressive behavior, predation, social learning, startle response, mating preference, and shoaling, which represents the complex interaction of fishes moving together. These features emphasize medaka as an emerging model for neurobehavioral research [3]. Moreover, due to their high genetic resemblance to humans, their genetic examination progresses rapidly, opening up new approaches for studying the genetic control of behavior and establishing their role as a highly valuable human disorders study model [12,13,14,15,16,17]. Additionally, medaka also affords many beneficial traits for behavioral studies, including its central nervous system (CNS), which has a basic structure that resembles those of amniotes [18]. Finally, yet importantly, medaka also can provide invaluable data for comparative research on zebrafish [19]. However, the current behavioral profiles based on medaka mainly focus on their early stages, or only involve simple tests in aquatic toxicology research or ecotoxicological risk assessment [10]. Therefore, systemically comparing their behavioral features from simple to complex tasks is required to obtain full toxicity information of several toxicants.

Many animals exhibit inter- and intra-specific behavioral diversities, generated not only by genetic factors, but also by learning and development [20,21]. These behavioral diversities in some animal groups are thought to be a significant factor contributing to the emergence of social organization, influencing fitness, and are suggested to be under natural selection [22]. Thus, in evolutionary ecology, this type of biological variation in behavioral characteristics has attracted the interest of many researchers [18,23,24]. These variations are widely recognized in both clinical populations and animal models of human disorders. It is prevalent for the CNS phenotypes, such as neuropharmacological, behavioral, and toxic responses [25,26]. However, the reasons causing the individual differences that are maintained in natural populations, and the connection between genetic polymorphisms and behavioral diversities, remain unsolved. While interspecific behavioral diversities have been relatively well studied in rodents and primates, aquatic models are far less characterized in this case [27]. Initially, fish behavior was considered ‘simple’ and instinctive; however, it is recognized as homologous to mammals, complex, adaptive, context-specific, and highly variable nowadays [28].

Unfortunately, even though medaka is a small freshwater fish that is commonly used as an animal model for aquatic toxicology research, it has received relatively little attention in behavioral studies, specifically in interspecific behavioral diversities, compared to zebrafish [10,29]. Thus far, few reports describe the differences in the behavioral traits among the medaka species and inbred strains, including a prior study by Hyodo-Taguchi that found some inbred strains of medaka tend to be attached to humans. In contrast, other inbred strains tend to avoid interactions with humans, and these responses may be mediated by a visual stimulus [30]. However, this difference has not been quantitatively analyzed based on paradigmatic experiments [18]. In addition, a prior study showed the great phenotypic diversity of *Oryzias woworae*, and studied the causative gene underlying divergence in sexually selected traits [31]. Later, based on the red pectoral fins that are unique to the male of this fish, *csf1* was revealed as a causative gene for red pectoral fins that can contribute to male reproductive success by the integration of genomic analysis and genome-editing technology. This result demonstrates that integrating genomic and phenomic approaches enables the identification of the causative genes underlying selected traits.

In the present study, we examine the interspecific variation in behavioral responses in adult medaka fishes with a series of behavioral assays to determine whether these fishes can be used in behavioral studies in a way similar to how zebrafish are used, or even in a more in-depth behavioral study considering the toughness of medaka over zebrafish in several extreme environmental conditions. Furthermore, biochemical studies were also carried out to help in elucidating the behavior results. We hypothesized that each species has its species-specific features, and the results of this study may provide information for other researchers to decide which medaka fish is more suitable for the study of behavioral responses, with a particular focus on neuroscience, pharmacology, and toxicology. Additionally, the present result can contribute to the growing database of phenotypical differences between several medaka fish species. The overview of the experimental design in the present study can be found in Figure A1.

## 2. Results

### 2.1. Novel Tank Assay Performance Comparison of Four Species of Medaka Fish

The novel tank assay is a test to inspect fish locomotor activity and its exploration ability to respond to the new environment [32]. Typically, fish spend most of the time at the bottom of the tank when they are introduced into a new environment, and they expand their swimming area to higher portions of the test tank after acclimating over time [33,34]. Four behavioral endpoints were used to observe their locomotor activity, which were average speed, freezing movement time ratio, swimming movement time ratio, and rapid movement time ratio. Meanwhile, regarding the exploratory behavior, the time in top duration, number of entries to the top, latency to enter the top, total distance traveled in the top, and average distance to the center of the tank (thigmotaxis) were quantified. During the test, each medaka group displayed a significantly different locomotor activity level to every other group. The highest level of locomotor activity was exhibited by *O. woworae* among the medaka groups. This phenomenon was shown by a high average speed, swimming, rapid movement time ratios, and a low level of freezing movement time ratio (Figure 1A–D). However, zebrafish still displayed higher locomotor activity than this medaka fish. Following *O. woworae*, a relatively high level of locomotor activity was also observed in *Oryzias dancena*. Even though not as high as *O. woworae*, *O. dancena* also displayed a higher average speed and rapid movement ratio than the other two medaka fishes (Figure 1A,D). Furthermore, a similar level of locomotor activity was detected in *O. latipes* and *O. sinensis*. While their average speed and rapid movement ratio were not statistically different from each other, different movement types were observed in these fishes, supported by the low level of freezing time movement ratio and high level of swimming time movement ratio of *O. latipes* over *Oryzias sinensis* (Figure 1B,C). Next, regarding the exploratory behavior, a different response to the novel environment from the zebrafish was displayed by all of the medaka groups. These differences were supported by the significant difference in all of the exploratory behavior-related endpoints between the zebrafish and all of the medaka fishes observed during the test (Figure 1E–I). Interestingly, *O. sinensis* showed a longer time in top duration than the other medaka fishes and even higher than the zebrafish (Figure 1E). However, a relatively short distance was traveled in the top indicating that this medaka swam slower than the zebrafish in the top portion of the test tank, which was also supported by a low level of average speed displayed by this fish (Figure 1A,I). Taken together, *O. woworae* possessed the highest locomotor activity among all of the medaka fishes, while the zebrafish locomotor activity was still higher than that of the medaka fish. Also, each medaka fish showed a specific and unique exploratory behavior, and this behavior was significantly different from the zebrafish. The detailed statistical analysis results of this test can be found in Table A1.

### 2.2. Aggressiveness Comparison of Four Species of Medaka Fish

Next, to evaluate the aggressiveness level of the fishes, a mirror biting assay was carried out. This measurement was performed by counting the relative interaction time of the fish with its mirror-reflecting image. Generally, to drive away from the potential intruder, fish immediately display mirror biting behavior when introduced into a tank with a mirror [35]. Similar to the novel tank test result, each medaka fish displayed a different level of aggressiveness. The most pronounced aggressive behavior was shown by *O. sinensis*, followed by *O. woworae*. This finding was indicated by a significantly higher mirror biting time percentage and the longest duration on the mirror side percentage of these two medaka fishes than the other two medaka fishes, *O. dancena* and *O. latipes* (Figure 2A,B). Interestingly, their level of aggressiveness was also found to be significantly higher than *Danio rerio*. On the other hand, *O. dancena* and *O. latipes* displayed a comparable level regarding their aggressiveness, which was also similar to *D. rerio* (Figure 2A,B). The detailed statistical analysis results of this test can be found in Table A2.

### 2.3. Comparison of Predator Avoidance Test Performance for Four Species of Medaka Fish

Afterward, we evaluated the fish’s fear level when facing their predators in the predator avoidance test [36]. The convict cichlid (*Amatitlania nigrofasciata*) was used in this study as a stimulus fish to promote the fear level of the tested fish based on the previous protocol [37]. The least predator avoidance behavior was displayed by *O. dancena*, followed by *O. latipes*. A high level of approaching predator time percentage and a low level of the average distance to the predator’s separator exhibited by these medaka fishes during the test indicated that they were not as fearful as the other fishes, including the zebrafish (Figure 2C,D). However, *O. woworae* and *O. sinensis* still displayed quite clear predator avoidance behavior on a similar level with the zebrafish. The detailed statistical analysis results of this test can be found in Table A2.

### 2.4. Conspecific Social Interaction Comparison of Four Species of Medaka Fish

Later, a conspecific social interaction test based on a similar rodent paradigm was conducted to evaluate the fishes social behaviors. This test is conducted by observing their interactions with the conspecifics, and it is one of the useful assays to study fish social phenotypes [35]. From the results, *O. latipes* displayed the slightest interest to interact with their conspecific. This phenomenon was supported by a low level of conspecific interaction time percentage and the longest conspecific interaction percentage, and a high level of average distance to the conspecific separator (Figure 3A–C). Following *O. latipes*, *O. dancena* and *O. sinensis* showed a slightly more pronounced conspecific social interaction during the test. The most profound conspecific interaction among the tested medaka fishes was observed in the *O. woworae* group. In addition, this medaka fish also exhibited a comparable level of this social behavior with the *D. rerio*. The detailed statistical analysis results of this test can be found in Table A2.

### 2.5. Comparison of Shoaling Behavior for Four Species of Medaka Fish

Shoaling, an innate behavior for several fish to swim together, was observed in each medaka fish. Generally, this behavior was intended to reduce anxiety and the risk of being captured by the predators [38,39]. *O. latipes* and *O. dancena* formed a quite tight shoal during the test, shown by the low levels of all of the behavioral endpoints, which are the average inter-fish distance, average shoal area, average nearest neighbor distance, and average farthest neighbor distance (Figure 4A–D). Meanwhile, *O. woworae* and *O. sinensis* displayed a slightly looser shoal, which is interesting since a similar shoal size was also observed in *D. rerio*. The detailed statistical analysis results of this test can be found in Table A2.

### 2.6. Comparison of Circadian Locomotor Activity Rhythms for Four Medaka Species

Next, we assessed their circadian locomotor activity rhythm since this daily rhythm of gross locomotor activity is frequently used as an assay of the circadian rhythmicity of animals, especially mammals [40]. From the results, all the medaka fishes displayed a significantly different circadian locomotor activity pattern compared to the zebrafish (Figure 5A). The high levels of locomotor activity showed these differences during both the day and night cycles. Meanwhile, regarding the medaka fish results, differences between each species were also observed during the experiment. In the day cycle, the highest locomotor activity was exhibited by *O. sinensis*, followed by *O. dancena*, *O. latipes*, and *O. sinensis*, which was consistent with the novel tank test results, shown by the differences in average speed, average angular velocity, freezing, swimming, and rapid movement time ratios (Figure 5B–G). Also, in terms of movement orientation, more pronounced zig-zag-like movement was shown by every medaka fish compared to the zebrafish. This phenomenon was shown by the high level of meandering measured during the day cycle, which was also observed during the night cycle (Figure 5J). Furthermore, similar results regarding locomotor activity were also found in the night cycle, where *O. woworae* possessed the highest locomotor activity among the rest of the medaka fishes, indicated by the differences in the average speed, average angular velocity, freezing, swimming, and rapid movement time ratios (Figure 5H,I,K–M). The detailed statistical analysis results of this test can be found in Table A2.

### 2.7. Biochemical Assay of Biomarker Expression in the Brain

Since behavior variations between each medaka fish were observed in previous behavioral tests, it was intriguing to investigate the contents of several neurotransmitters, the antioxidant activity, and oxidative stress that might play roles in these differences. Interestingly, based on the neurotransmitter levels, we found that *O. woworae* and *O. dancena* showed significantly higher levels of all of the tested neurotransmitters (serotonin (5-HT), dopamine (DA), and norepinephrine (NE)), except for acetylcholinesterase (AChE), than *O. latipes* and *O. sinensis*. Furthermore, a similar result was also observed in the stress hormone cortisol (Table 1). On the other hand, while *O. woworae* still showed the highest level of AChE, *O. dancena* possessed a comparable level of this neurotransmitter to *O. sinensis*, followed by *O. latipes* that also displayed a similar level of AChE to *O. sinensis*. In addition, this pattern was also found in the oxidative stress-related markers, which are catalase (CAT) and reactive oxygen species (ROS) (Table 1). The detailed statistical analysis results of this test can be found in Table A3.

### 2.8. PCA Analysis and Hierarchical Clustering Analysis of Several Medaka Fish Behavioral Endpoints

To explore the behavioral phenomics between several different medaka fish species, principal component analysis (PCA), hierarchical clustering, and heatmap comparison were performed after all of the behavioral tests. This process is also vital to reduce the data dimension and complexity. As the outgroup, the AB strain zebrafish (*D. rerio*) behavioral data were included to conduct a more profound study about the behavioral pattern differences between the tested medaka fishes. Two major clusters were generated from the hierarchical clustering result, separating *O. latipes*, *O. dancena*, and *O. sinensis* in one cluster, with *O. woworae* and *D. rerio* in another cluster (Figure 6A,B). The definition of all of the behavioral endpoints was described in Appendix A
Table A4.

### 2.9. Phylogenetic Relationships of Four Medaka Species

To verify the relationship between each medaka fish species tested, we constructed a phylogenetic tree of medaka and *D. rerio* as the outgroup based on the NADH dehydrogenase subunit two gene (*nd2*). The medaka fish were classified into three groups from the phylogenetic tree (Figure 7), including *celebensis*, *javanicus*, and *latipes* species groups, as described by Kinoshita et al. [9].

## 3. Discussion

This is the first study to demonstrate the comparison of the innate behaviors between several medaka fish species to the best of our knowledge. As we predicted, each species has its species-specific features in each of the behaviors tested. *O. woworae* displayed the highest locomotor activity, while each medaka exhibited a unique exploratory behavior to each other. Moreover, this phenomenon was also obviously shown in the circadian locomotor activity rhythm test. Furthermore, this fish, together with *O. sinensis*, exhibited a significantly high aggressiveness level than *D. rerio*. Meanwhile, regarding predator avoidance behavior, these fishes showed similar fear-like behavior to *D. rerio*, while this behavior response to the predator was not clearly observed in *O. dancena* and *O. latipes*. *O. woworae* also showed a more distinguished social behavior in the same manner with *D. rerio* than other medaka fishes. Lastly, while all of the medaka fishes showed interest in forming a shoal with their conspecifics, a quite tight shoal was displayed by *O. dancena* and *O. latipes*.

The novel tank test has been widely used to study animal habituation. Generally, habituation is defined as a change in locomotor activity and exploratory behavior over time to sustain an animal’s survivability [41]. Based on the prior study, this behavior test on the zebrafish provided results comparable to other similar studies in other animal species, including medaka [42]. The typical behavioral pattern changes of all of the medaka, in a stereotypical manner, were demonstrated by the current results. This phenomenon indicated that the animals became familiar with the novel location as the exposure time increased. Consistent with this result, a previous study demonstrated a similar pattern change during the habituation of *O. latipes* in an open-field test [19]. In terms of their differences from the zebrafish, all of the medaka fishes exhibited a relatively lower locomotor activity than the zebrafish most of the time. This result is somewhat similar to a prior study in another species of medaka (*Oryzias javanicus*), which found that the medaka fish were passive compared to the *D. rerio* who were more active and aggressive [43]. Moreover, a high thigmotaxis level, a preference of animals towards the periphery of a novel arena and avoiding the center area, during the test was displayed by all of the medaka fish [44]. This result is plausible since zebrafish tend to spend time in the center area of a tank after it is habituated, while this behavior is not shown in medaka [45]. All of the medaka fishes displayed similar phases of habituation in the novel environment. This finding confirmed the usefulness of medaka and novel tank tests to investigate the habituation phenotype even though each species possessed different degrees of locomotor activity and exploratory behavior.

Similar to zebrafish, mirror approaching behavior in medaka is a well-established fish paradigm that reflects their aggressive behavior [46]. When a single medaka is placed in a tank, it swims freely in all directions; however, the same individual will tend to swim close to the mirror when a mirror is placed on one side of the tank [47]. Unfortunately, while this paradigm is well studied in medaka, most of the studies were done in *O. latipes* [3,47]. Thus, this is the first study that revealed the mirror approaching behavior in other medaka species. Surprisingly, in this study, while *O. latipes* displayed a similar mirror approaching behavior to *D. rerio*, other medaka fishes exhibited more robust mirror approaching behaviors. Since mirror approaching behavior in medaka is considered a simple and robust model of socially induced anxiety, the high level of mirror biting time might indicate a higher stress level in these fishes than *O. latipes* and *D. rerio*. Interestingly, cortisol, one of the primary fish hormones, was found to be lower in *O. latipes* compared to other medaka fishes, especially *O. dancena* and *O. latipes*. These observations indicate that cortisol might affect the differences in their aggressive behavior since it is involved in the medaka fish stress response [48]. Also, this speculation was based on the prior study in diazepam and fluoxetine, which are anxiolytic agents that decreased the mirror biting time and the stress response measured by the cortisol level [3,47,49]. Furthermore, based on the ELISA results on ROS, the molecules responsible for the signaling stress response, we hypothesize that different basal levels of ROS might also influence the differences in their behavior. This possibility was based on the previous research in adult zebrafish that showed anxiety-like or stress-like behavior, which was likely contributed to by higher ROS levels [50,51]. Besides ROS, this speculation is also supported by the variation in the catalase basal levels of each medaka fish. Catalase is primarily a peroxisomal enzyme that catalyzes the enzymatic decomposition of H_2_O_2_, and in rats it is associated with depression-like behavior associated with Alzheimer’s disease improvement [52,53]. Lastly, aggression also has been linked to serotonergic function in a variety of invertebrate and vertebrate species. Generally, a high level of serotonin activity clearly shown in the results is associated with low levels of aggressive behavior [54].

Avoiding predation is an essential behavioral reaction critical for survival, and may have a significant fitness component. Many animals, including fish, sense a danger of predation through a multitude of methods, such as alarm signals, calls, and chemical cues. To avoid predation, the animals have to respond to stimuli that represent the presence of danger properly [36,55]. It not only helps the researchers to elucidate the evolution and ecology of the studied species, but analysis of such fear responses may have clinical relevance [56]. Unfortunately, regardless of the promising advantages, only a few studies of this predation response were done in medaka. Moreover, most of these studies only used *O. latipes* as an animal model [55,57]. However, in the current study, we found that each medaka species responded to the presence of *A. nigrofasciata* as the fear stimulus in different magnitudes. Interestingly, while the *O. latipes* fear responses were clearly demonstrated in a prior study, this behavior was not shown in this experiment. This difference might be due to the different stimuli used to elicit the alarm response. While we stimulated the fear response by exploiting the visual cue from another fish, the prior study used a conspecific skin extract as the chemical cue of alarm substance [55]. Thus, each medaka fish might respond differently to other fish species or a different stimulus upon their first exposure to these fish, which needs to be confirmed in future studies. In addition, a more robust reaction in response to the predator stimulus was observed in *O. woworae* and *O. sinensis*. These results might indicate that these medaka fishes displayed a fear response in the presence of the stimulus since their response was similar to *D. rerio*, which is already well studied and proven to elicit a fear response in a similar situation [37,58,59]. One possibility that caused the differences is related to the specific basal serotonin content of each medaka fish, as shown in the current study. As mentioned above, serotonin is a good physiological indicator of various types of stress in fish. Mosienko et al. demonstrated that serotonin deficiency in certain regions of the brain, exhibited by *tph2*-knockout medaka, was associated with elevated stress and fear-related behaviors [60].

Generally, the development of social behavior is observed among various vertebrates, from fish to mammals, and coordinated with social factors such as the presence of conspecific, physiologic, and environmental factors [61,62]. In medaka fish, especially *O. latipes*, this social behavior had been well addressed previously. A prior study discovered that medaka fish robustly and reproducibly maintained proximity to a single target conspecific fish since it is attracted by conspecifics’ biological motion [63,64]. However, whether other medaka species also possess this social behavior by the presence of conspecific remains unknown. Even though *O. latipes* was found to maintain proximity to its conspecific, other medaka species also displayed a similar behavior, even in a higher magnitude and more robust manner. These results are intriguing since some of these medaka species, especially *O. woworae*, displayed social behavior as clear as *D. rerio*, a well-established animal model for this behavior [37,65]. Furthermore, shoaling, a simple form of affective behavior displayed in social fish species, is usually observed in small fish, including medaka (*O. latipes*) and zebrafish. This behavior may directly influence the survival of individuals and future population structures [11]. In the previous study, medaka was demonstrated to form shoals only with conspecifics and not with other species [66]. In this study, all medaka fishes formed a shoal that was even tighter than *D. rerio* for some medaka species, which were *O. dancena* and *O. latipes*. These results are plausible since medaka are known to have high visual acuity and exhibit a strong tendency to form shoals [67]. The differences in the social behaviors of each medaka fish might be related to the differences in their dopamine and norepinephrine levels, as shown in the ELISA results. These neurotransmitters are involved in the organization of the stress response in vertebrates, and together with serotonin, they are involved in the mediation of behavioral and neuroendocrine responses to social stress [68,69]. A prior study in teleost shows that the brain dopamine level increased after being stressed. Thus, the current results indicate that some medaka fishes might have different levels of stress compared to the others, which is consistent with the mirror biting test results. The stress is plausible since recent research discovered that grouping and pairing are more stressful to male medaka than isolation [48]. Regarding the norepinephrine content, the previous finding found that juvenile lake sturgeon (*Acipenser fulvescens*) held in isolation had a significantly longer norepinephrine response than fish held with conspecifics [70]. In addition, another study in zebrafish found that the whole-brain dopamine level was also found to be associated with the development of shoaling [71]. In a prior study, the dopamine level was found to have rapidly risen in the brain of adult zebrafish in response to social stimuli [72].

The time of oviposition in medaka fish, especially in *O. latipes*, was well-accounted several years ago. Interestingly, this rhythm is controlled by a circadian pacemaker entrained by light–dark (LD) cycles since it was gradually disappeared under continuous light [73]. The locomotor activity rhythm seems to be dependent on the light intensity during dark periods [74]. Here, typical diurnal circadian locomotor activity rhythms were displayed by all of the medaka fishes. Even though they were not as pronounced as in *D. rerio*, they are consistent with their natural behaviors [75]. This phenomenon might be explained by the circadian rhythm trait of medaka fish, especially *O. latipes*, which can be entrained by feeding time [76]. Thus, the feeding schedule that occurred during the day cycle in the present study might play a role in affecting this behavior since a prior study demonstrated that this entrainable rhythm is shown to be a persistent behavior. In their study, it was discovered that after a three-day fast, a meal-feeding readily entrained circadian rhythm of agonistic behavior of *O. latipes* remained fixed to the feeding time [77]. However, the reader should not conclude that all medaka species are capable of being entrained to feeding. Thus, future studies are needed to be conducted to verify this speculation. Furthermore, *O. woworae* was found to be the most active fish regarding their locomotor activity levels compared to other medaka fishes, which is consistent with the novel tank test results. This phenomenon could be related to the high level of serotonin and AChE measured in the fish. This speculation was taken since serotonin has been reported to play a positive role in regulating locomotor activity in many animals, while AChE is an enzyme responsible for the breakdown of acetylcholine, which plays an important role in modulating neuromuscular activity in neural synapses [78]. This trend is also observed in other medaka fishes.

In summary, the behavioral differences between each medaka fish might be associated with their different basal level of neurotransmitters. This phenomenon has been briefly noted in a previous study. However, the comparison only covered three kingdoms of living organisms [79]. Nevertheless, even though this difference was not fully investigated, it has been mentioned in several prior studies, including in fish. In normoxia conditions, three fish species, goldfish, tilapia, and carp, possessed different energy status levels, lactic acid accumulation, and amino acid patterns in brain tissue [80]. In addition, by using a combination of immunohistochemistry and confocal imaging, a previous experiment confirmed the differences in the serotonin and acetylcholine contents in the gill’s filament and lamellae of various fish species, including goldfish (*Carasius auratus*), sockeye salmon (*Oncorhynchus nerka*), trout, and *O. latipes* [81]. Furthermore, several prior findings also found some variations in the levels of adrenaline and noradrenaline stored within the chromaffin tissue of various species of fish, such as cyclostomes, dipnoans, elasmobranchs, ganoids, and teleosts [82]. Lastly, Aprison et al. also found some differences regarding glycine concentration, an amino acid that has potent inhibitory actions on the vertebrate nervous system, at various levels of the neuraxis in five different vertebrates [83]. Additionally, these phenomena might be related to the different genes underlying the divergences in selected traits between species, particularly in vertebrates, which needs further studies to confirm this hypothesis [84]. Overall, this is the first study comparing the basal levels of neurotransmitters between each species of medaka fishes to the best of our knowledge. Thus, further studies are required to deeply investigate this phenomenon.

Next, the phenomic-based hierarchical cluster generated two major clusters, which were *O. latipes*, *O. dancena*, and *O. sinensis* in one cluster, and *O. woworae* and *D. rerio* in another cluster (Figure 6A,B). In the first cluster, *O. latipes* and *O. dancena* were closer to each other than *O. sinensis*. This categorization was plausible since these medaka fishes exhibited several behaviors that were not observed in other fish, such as less pronounced predator avoidance behavior (endpoint 3-1) and tightened shoal formation (endpoint 5-1 to 5-4). Meanwhile, even though *O. sinensis* was found to be in this cluster, it showed a different behavioral pattern to the two medaka fishes in several vital endpoints. The most evident difference was in one of the novel tank test endpoints, the time in top duration (endpoint 1-1-5 and 1-2-5). As already mentioned in Figure 1, a significantly high level of this endpoint was displayed by this medaka fish during the whole 30 min of the novel tank test. Next, it was found that *O. woworae* and *D. rerio* belonged in the same cluster. After further investigation, we discovered that these two fishes exhibited similar behavior patterns, especially in the locomotor activity-related endpoints, such as the average speed and rapid movement ratio. A strong conspecific interaction shown by these fishes also became one of the factors that defined this grouping. Afterward, to precisely determine the phylogenetic position of each medaka fish and *D. rerio*, we estimated the phylogenetic relationship using published whole-genome datasets as references. The tree indicates that *O. sinensis*, *O. latipes*, and *O. dancena* formed a monophyletic group, which is consistent with previous trees based on a concatenated mitochondrial sequence matrix [85]. Interestingly, this genetic-based grouping follows with the salinity tolerances of these fishes, which are euryhaline for *O. dancena*, *O. latipes*, and *O. sinensis* [86,87,88], and stenohaline for *O. woworae* and *D. rerio* [89,90]. This result is plausible since *O. sinensis* is described as the subspecies *O. latipes sinensis* Chen. These species have approximately the same length of anal–fin rays and chromosome arms numbering 58 or more, distinguishing them from putative close relatives [88,91]. Furthermore, *O. dancena* and *O. woworae* were phylogenetically clearly separated from *O. sinensis* and *O. latipes*, which is also in agreement with other phylogenetic trees from previous studies that are based on the nuclear *tyrosinase*, mitochondrial 12S and 16s rRNA genes, and several characters. Based on these studies, the *Oryzias* species itself is ramified into three monophyletic groups. While *O. dancena* belongs to the *javanicus* clade, both *O. sinensis* and *O. latipes* belong to the *latipes* clade, which can elucidate the phylogenetic tree result of the present study. The *javanicus* clade is a monoarmed chromosome group and possesses subtelo- and acro-centric chromosomes that distinguish it from the other clades [9,85,88,92]. Next, as reported in a phylogeny based on the 3440-bp concatenated mitochondrial and nuclear sequences by Mokodongan and Yamahira, *O. woworae* was genetically distinct from other medaka fishes, such as *O. latipes* and *O. javanicus*. This grouping might also be caused by the different wider salinity tolerance between the medaka fishes. While medaka fishes, including *O. latipes*, *O. sinensis*, and *O. dancena*, have a wide salinity tolerance, this trait was not possessed by *O. woworae*, which might also explain the close distance between this fish and *D. rerio* in the phylogenetic tree. However, based on the preliminary phylogenomic analyses, the distance between zebrafish and medaka is due to the differences in the 20 nuclear protein-coding genes [93,94,95,96]. Interestingly, the phylogenetic tree and phenomic-based hierarchical cluster generated in this study displayed several resemblances, including the interspecies relationship between zebrafish and the medaka fishes. However, a slightly different result to the phylogenetic tree was observed in the medaka interspecies comparisons. The phenomic-based hierarchical clustering result showed that *O. latipes* and *O. dancena* appeared to be more closely related to each other than either of them was to *O. sinensis*. Thus, this is evidence that genetic-based phylogenetic analyses might have a different relationship among the species to the phenomic-based cluster analysis. However, a prior study has already addressed those differences regarding the relationship between zebrafish, medaka, pufferfish, and cichlids. In their study, the molecular data supported a close relationship between atherinomorphs (including the medaka and platy) and putatively more derived perform fish such as the cichlids. At the same time, previously the Atherinomorpha have been historically placed in an intermediate position among the other branches of the acanthomorph tree since they share several putative ‘primitive’ morphological features with more basal teleosts. Thus, their phylogenomic analysis revealed an unexpected relationship among the other three species, contrary to traditionally held systematic views based on morphology [96].

Next, to observe their reproducibility, we also calculated the coefficient of variation of each medaka fish species and *D. rerio* from every behavioral test. Based on the result in Table A5, *O. woworae* displayed the lowest average coefficient of variation between all of the medaka fishes, even though it was still higher than *D. rerio*. This result indicates the stronger reproducibility of *O. woworae* than other medaka fish in these behavior tests. A high coefficient of variation may impact the animal usage of an experiment since this condition is a problem to obtain significant results. Thus, a large sample size has become mandatory to overcome this problem. Moreover, this problem also means that the researchers might face ethical-related issues [29,97].

## 4. Materials and Methods

### 4.1. Animal Husbandry

Indian ricefish (*Oryzias dancena*) and Daisy’s ricefish (*Oryzias woworae*) were purchased from a local pet store, while Chinese ricefish (*Oryzias sinensis*) and Japanese ricefish (*Oryzias latipes*) were obtained from the Freshwater Bioresource Center at National Chiayi University. The AB strain zebrafish were obtained from the Taiwan Zebrafish Core Facility at Academia Sinica (http://icob.sinica.edu.tw/tzcas/, accessed on 26 November 2020). All fish in each experiment were mixed gender of 6–10 months old adults in healthy condition. One month prior to the experiment, all tested fish species were reared in the centralized fish facility with a recirculating aquatic system at 28 ± 1 °C and a 10/14-h dark/light cycle. This process was important to eliminate some external factors that might affect their behavior performance, such as the stressful condition during the transfer [98]. The conductivity of the circulating system’s water was kept between 300–1500 µS with pH 7.0–7.5. Ultraviolet (UV) light was utilized to filter the water constantly. All fish were held and raised in a trapezoid plastic tank with 34 cm at the top, 23 cm along the bottom, 19 cm along the diagonal side, 18 cm high, and 27 cm wide filled with 8 L of filtered water. Feed was given twice a day (09:00 and 17:00) with either lab-grown brine shrimp or commercial dry food. The general maintenance procedures and housing conditions were as previously described by Avdesh et al. [99].

### 4.2. Animal Ethics and Behavioral Tests

All fish experiments were performed following the guidelines issued by the Institutional Animal Care and Use Committees (IACUCs) of Chung Yuan Christian University (application number: CYCU106025, issue date 6 May 2018). All behavioral tests were conducted within the morning until afternoon (10:00 to 16:00), except for circadian locomotor activity rhythm, in a temperature-controlled room (26 ± 1 °C). After the acclimation process, a battery of behavioral tests, which were the novel tank, mirror biting, predator avoidance, conspecific social interaction, and shoaling tests based on the previous method, was conducted in all the fish groups [37]. A total number of 142 fish were used in behavior tests (*n* = 30 for *D. rerio*, *O. dancena*, *O. latipes*, and *O. woworae*; *n* = 22 for *O. sinensis*). The slightly lower number of *O. sinensis* used compared to other fishes was caused by the low availability of this fish in our region during the time period. In addition, the differences in sample size number of fish groups in several behavior tests were due to the unknown death of the fish during the rest period between the behavior tests. The entire tests were done using a trapezoid tank with 22 cm along the bottom, 28 cm at the top, 15.2 cm high, and 15.9 cm along the diagonal side, and later, this tank was filled with ~1.25 L of circulating filtered water. In avoiding stress conditions in the tested fish, all the behavior tests, excluding the circadian rhythm locomotor activity test, were divided into three sessions and carried out on a single week. The first session consisted of the novel tank and shoaling tests. A novel tank test was conducted as the first behavior test since the tested fish had not been introduced to the test tank previously, thus, their behavioral responses toward the novel environment would be more genuine and reliable. The mirror biting and social interaction tests belonged to the second session, followed by the predator avoidance test in the last session since in this test, the convict cichlid (*Amatitlania nigrofasciata*) was supposed to induce the fear response on the tested fish and since fear can be possessed by the fish for some period, it could affect their behavior in the following behavior tests. Before all of the tests, except the novel tank test, 3–5 min of acclimation in the test tank was applied to all tested fishes. Afterward, a 5-min video recording was conducted. Meanwhile, in the novel tank test, fish behavior was recorded for 1 min as soon as the fish were exposed to the test tank. The behaviors were recorded at 5 min intervals for 31 min. Next, the behavior tests were continued with the circadian locomotor activity rhythm test on the following week, which monitored its locomotor activity for 24 h [100]. This test was conducted in a 30 × 30 × 7.5 cm acrylic tank filled with ~3 L of filtered water. A lightbox, which consisted of two types of a light source (chip-on-board (COB) light-emitting diode (LED) and 940 nm infrared LED), was placed below the tank. Later, the fish locomotion during 24-h test was recorded for 1 min every hour. Next, idTracker (http://www.idtracker.es/, accessed on 25 May 2020), open-source software, was used to collect and convert the fish movement data to trajectories from all of the recorded videos [101]. All fish behavior tests were done in triplicate.

### 4.3. Brain Tissue Preparation, Total Protein Determination, and Quantification of Neurotransmitters, Stress Hormones, and Oxidative Stress Markers

In obtaining the brain tissues, immediate anesthesia and euthanasia were performed in medaka fishes by immersing them in tricaine solution (A5040, Sigma, St. Louis, MO, USA). Later, biochemical analyses were conducted on their whole-brain tissue extract. Ice-cold phosphate-buffered saline (PBS) in volumes of 10 (*v*/*w*) was used to standardize a single homogenate of two to three whole medaka fish brains at pH 7.2. Afterward, to homogenize the tissue a bullet blender (Next Advance, Inc., Troy, NY, USA) was utilized. After 15 min of 13,000 rpm centrifugation, the supernatant was transferred to a sterilized microtube and stored at −20 °C. Subsequently, Pierce BCA protein assay kit (23225, Thermo Fisher Scientific, Massachusetts, MA, USA) was applied to measure the total brain tissue’s protein level. After the color was formed, it was analyzed by using a microplate reader (Multiskan GO, Thermo Fisher Scientific, Waltham, MA, USA) at 562 nm. Subsequently, all medaka fish brain tissues were analyzed to compare the differences in neurotransmitters, stress hormone, and oxidative stress marker levels between each species of the medaka fishes. Several neurotransmitters, including serotonin (5-HT, ZGB-E1572), acetylcholine esterase (AChE, ZGB-E1637), dopamine (DA, ZGB-E1573), and norepinephrine (ZGB-E1571), were measured by target-specific ELISA kits. Meanwhile, tissue oxidative and anti-oxidative stress markers, reactive oxygen species (ROS) and catalase (CAT), respectively, and cortisol, one of the stress hormones, were also quantified by ELISA kits (ZGB-E1561, ZGB-E1598, and ZGB-E1575 Zgenebio Inc., Taipei, Taiwan). Later, a microplate reader (Multiskan GO, Thermo Fisher Scientific, Waltham, MA, USA) was utilized to measure the absorbance at 450 nm and compared it to the standard curve to quantify the relative concentration of the target protein. Ten biological and three technical replicates were applied in the analysis (*n* = 30, except *O. latipes* (*n* = 29)).

### 4.4. Statistical Analyses

In determining the statistical difference between each group, statistical analyses were carried out using GraphPad Prism (GraphPad Software version 8 Inc., La Jolla, CA, USA). For all behavioral and biochemical data analyses except the novel tank test, Kruskal–Wallis with uncorrected Dunn’s test was conducted to find the statistical differences in each group with every other group since the data are not normally distributed [102]. Meanwhile, two-way ANOVA with Geisser–Greenhouse correction continued, and uncorrected Fisher’s LSD test was used for the novel tank test. Data for each fish group are expressed as either median with interquartile range, the mean with a 95% confidence interval (CI), or standard deviation (SD). The statistic details for each behavioral and biochemical test are summarized in Table A2 and Table A3, respectively.

### 4.5. PCA, Heatmap, and Clustering Analysis

All the behavioral endpoint values from all the tested fish in every behavior test were input into a comma-separated values type file (.csv) using Microsoft Excel. All of the essential behavioral endpoints based on the prior study are listed and explained in Table A4 [37]. Next, the .csv file was uploaded to ClustVis (https://biit.cs.ut.ee/clustvis, accessed on 19 November 2020), a web tool designed to visualize and cluster multivariate data. Later, unit variance scaling for each row was carried out in order to treat each variable equally. Furthermore, singular value decomposition (SVD) with the imputation method was used to calculate principal components since there were no missing values in the dataset [103].

### 4.6. Phylogenetic Tree Construction

The complete mitochondrial NADH dehydrogenase subunit 2 gene (*nd2*) sequences for each species of medaka fish were downloaded from the NCBI database with the accession numbers NC_012976 (*O. dancena*), NC_004387 (*O. latipes*), LC051726 (*O. woworae*), and NC_013434 (*O. sinensis*). The gene sequences of *nd2* of *D. rerio* (NC_002333), representing outgroup species, were also downloaded from the NCBI database. All nucleotide sequences *nd2* of all species described above were then aligned using Geneious software (Biomatters, Auckland, New Zealand). Subsequently, phylogenetic analysis was also conducted with Geneious software with default parameter settings.

## 5. Conclusions

To sum up, the current study demonstrated that the phenotypes of four medaka fish, which were *O. latipes, O. dancena*, *O. woworae*, and *O. sinensis*, were different in their behavior in various behavioral tests. These differences might be related to the different basal levels of several neurotransmitters, the stress hormone, and oxidative stress markers measured. Furthermore, while our phenomic-based hierarchical cluster results showed a similar interspecies relationship between zebrafish and medaka fishes to the phylogenetic tree, the interspecies comparison between each medaka fish of these two approaches demonstrated a slight difference in the relationship result between three medaka fishes. Our findings also suggest that many challenges remain to be addressed before robustly using the behavioral endpoints for ecological hazard evaluation, since the instability of some behavior performances can be observed in the medaka fishes. We believe that our results contribute to the growing database of phenotypical differences between several medaka fish species and provide one of the foundations for future phenomics studies of medaka fish. Moreover, comparative studies involving zebrafish and medaka are remarkably informative for identifying the highly conserved genetic control mechanisms [104,105]. Furthermore, considering the abundance of medaka fish strains listed in the National BioResource Project (NBRP) Medaka, it is intriguing to study the differences between each medaka strain in every aspect, including the relationship between their genetics, behavior, and biochemistry [106].

## Figures and Tables

**Figure 1 ijms-22-05686-f001:**
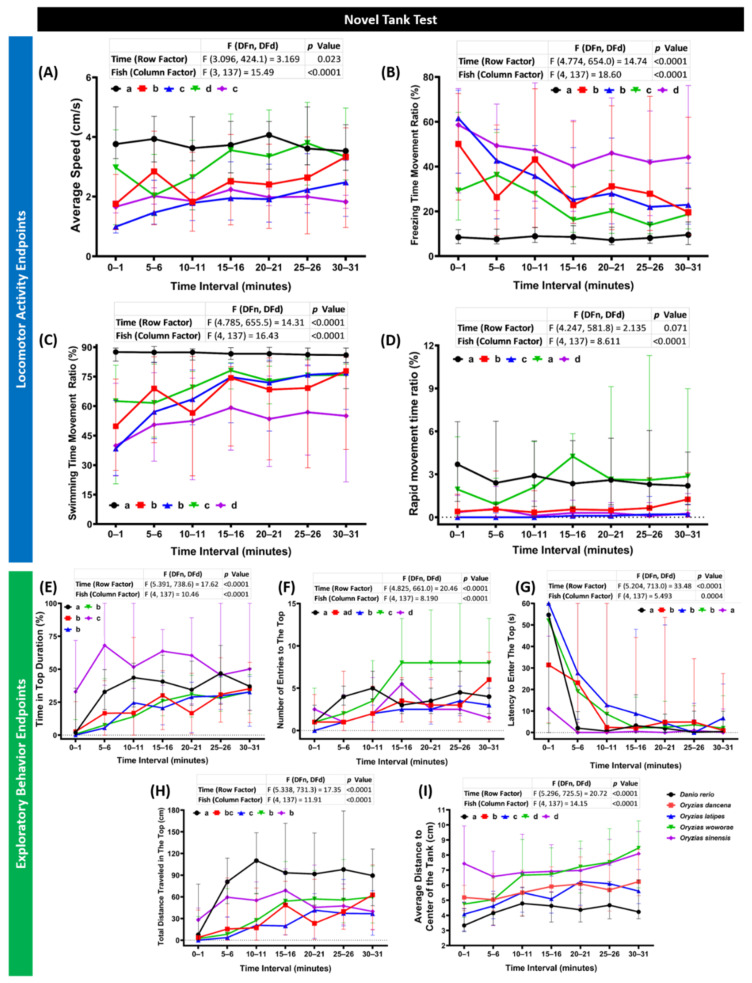
Novel tank behavior endpoints comparison between each tested fish species (medaka and zebrafish). (**A**) Average speed, (**B**) freezing time movement ratio, (**C**) swimming time movement ratio, (**D**) rapid movement time ratio, (**E**) time in top duration, (**F**) number of entries to the top, (**G**) latency to enter the top, (**H**) total distance traveled in the top, and (**I**) average distance to the center of the tank were analyzed. The data were analyzed by the two-way ANOVA test with Geisser–Greenhouse correction continued with uncorrected Fisher’s LSD test. Different letters (a, b, c, d) on the error bars represent a significant statistical difference (*p* < 0.05). The data are expressed as the median with interquartile range (*n* = 30 for zebrafish, *O. dancena*, *O. latipes*, and *O. woworae*; *n* = 22 for *O. sinensis*).

**Figure 2 ijms-22-05686-f002:**
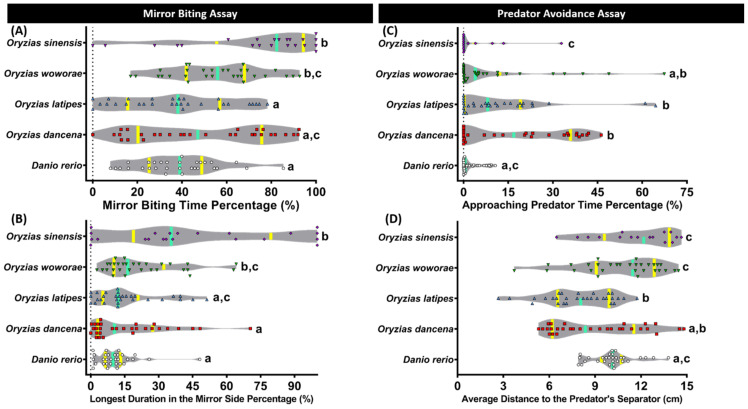
Mirror biting and predator avoidance behavior comparisons between each tested fish species (medaka and zebrafish). (**A**) Mirror biting time percentage and (**B**) longest duration on the mirror side percentage were analyzed in the mirror biting assay (*n* = 30 for zebrafish, *O. dancena*, *O. latipes*, and *O. woworae*; *n* = 22 for *O. sinensis*). (**C**) Approaching predator time percentage and (**D**) average distance to the predator’s separator were analyzed in the predator avoidance assay (*n* = 30 for zebrafish, *O. dancena*, *O. latipes*, and *O. woworae*; *n* = 16 for *O. sinensis*). The data were analyzed by the Kruskal–Wallis test continued with uncorrected Dunn’s test. Different letters (a, b, c) on the error bars represent a significant statistical difference (*p* < 0.05). The violin plot’s median and interquartile were labeled with the bold line colored with cyan and yellow.

**Figure 3 ijms-22-05686-f003:**
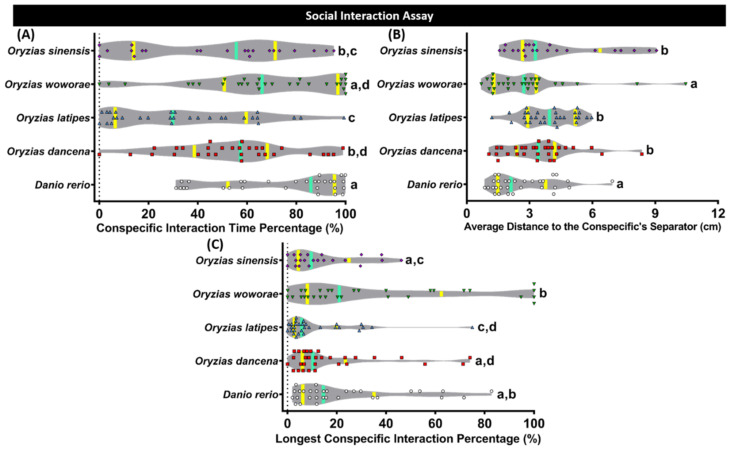
Conspecific social interaction behavior comparisons between each tested fish species (medaka and zebrafish). (**A**) Conspecific interaction time percentage, (**B**) average distance to the conspecifics separator, and (**C**) longest conspecific interaction percentage were analyzed. The data were analyzed by the Kruskal–Wallis test continued with uncorrected Dunn’s test. Different letters (a, b, c, d) on the error bars represent a significant statistical difference (*p* < 0.05) (*n* = 30 for zebrafish, *O. dancena*, *O. latipes*, and *O. woworae*; *n* = 21 for *O. sinensis*). The median and interquartile for the violin plot were labeled with the bold line colored with cyan and yellow.

**Figure 4 ijms-22-05686-f004:**
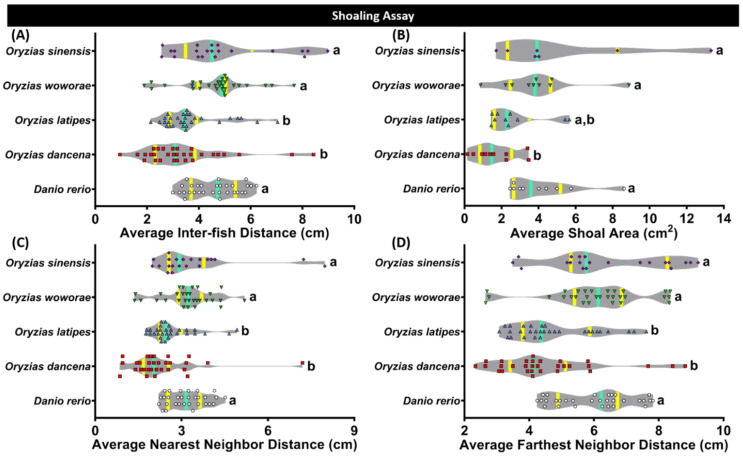
Shoaling behavior comparisons between each tested fish species (medaka and zebrafish). (**A**) Average inter-fish distance, (**B**) average shoal area, (**C**) average nearest neighbor distance, and (**D**) average farthest neighbor distance were analyzed. Groups of three fish were tested for shoaling behavior. The data were analyzed by the Kruskal–Wallis test continued with uncorrected Dunn’s test. Different letters (a, b) on the error bars represent a significant statistical difference (*p* < 0.05) (*n* = 30 for zebrafish, *O. dancena*, *O. latipes*, and *O. woworae*; *n* = 21 for *O. sinensis*). The median and interquartile for the violin plot were labeled with the bold line colored with cyan and yellow.

**Figure 5 ijms-22-05686-f005:**
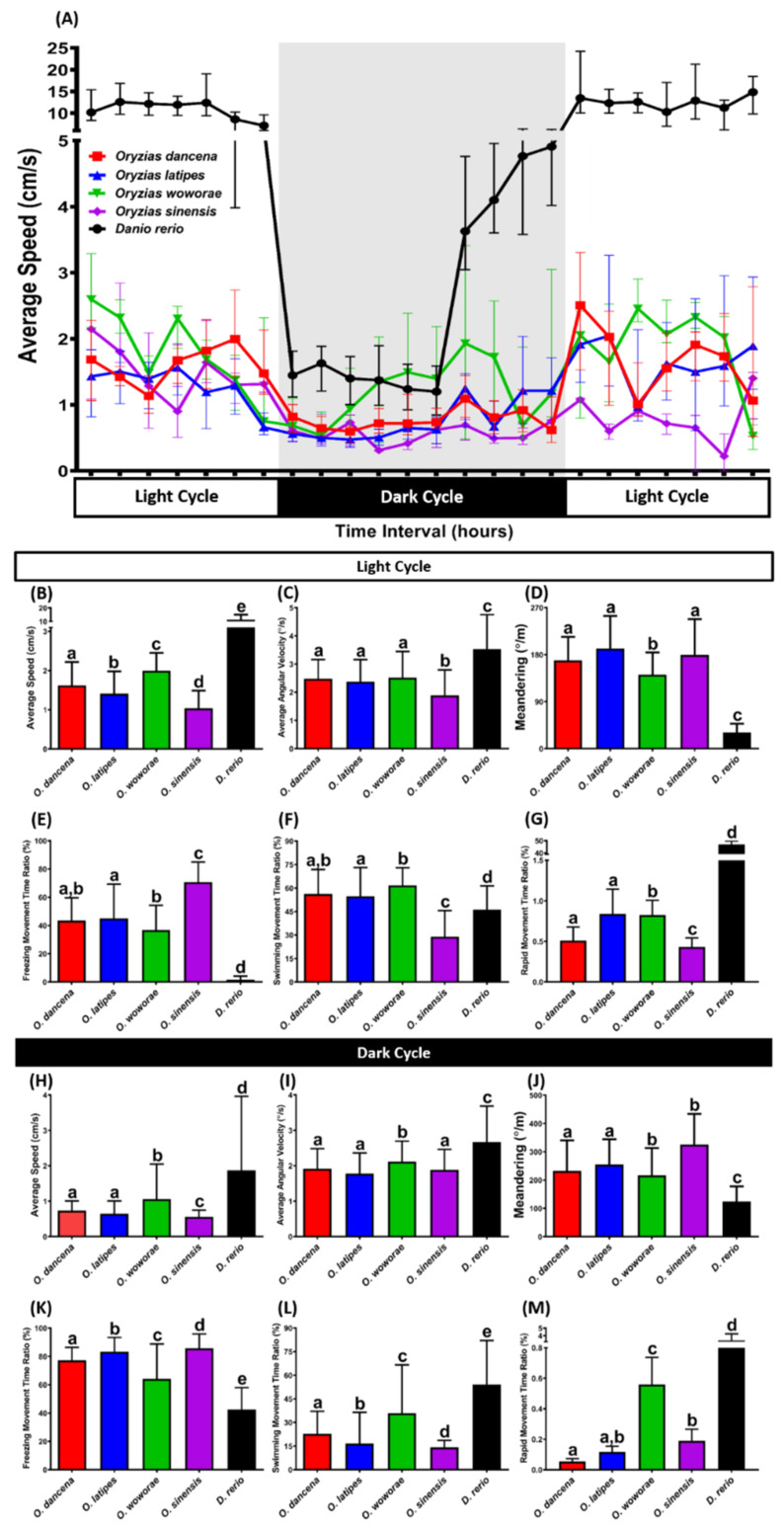
The circadian locomotor activity rhythm of medaka fish and AB strain zebrafish as the outgroup. (**A**) Comparison of the average speed between each tested fish species (medaka and zebrafish) during the day and night cycles. Comparisons of (**B**,**H**) average speed, (**C**,**I**) average angular velocity, (**D**,**J**) meandering, (**E**,**K**) freezing movement time ratio, (**F**,**L**) swimming movement time ratio, and (**G**,**M**) rapid movement time ratio between each tested fish species (medaka and zebrafish) in the day and night cycles, respectively. Data are presented as median with interquartile range, except for (**G**,**M**), which are presented as mean with a 95% confidence interval (CI) since some median of the data in those figures are 0. Data were analyzed by Kruskal–Wallis test continued with uncorrected Dunn’s test. Different letters (a, b, c, d, e) on the error bars represent a significant statistical difference (*p* < 0.05) (*n* = 18 for *O. dancena*, *O. latipes*, *O. woworae*, and *D. rerio*; *n* = 12 for *O. sinensis*).

**Figure 6 ijms-22-05686-f006:**
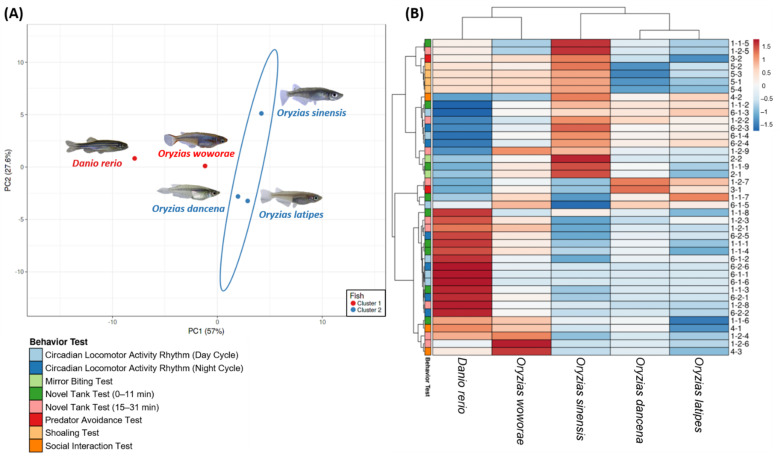
(**A**) Principal component analysis and (**B**) hierarchical clustering analysis of multiple behavior endpoints in several different medaka species and zebrafish. In (**A**), two major clusters from hierarchical clustering analysis results were marked with the red color (1st cluster) and blue (2nd cluster) circle. The behavioral data from zebrafish were also included as the outgroup to conduct a more in-depth study about their behavior differences pattern.

**Figure 7 ijms-22-05686-f007:**
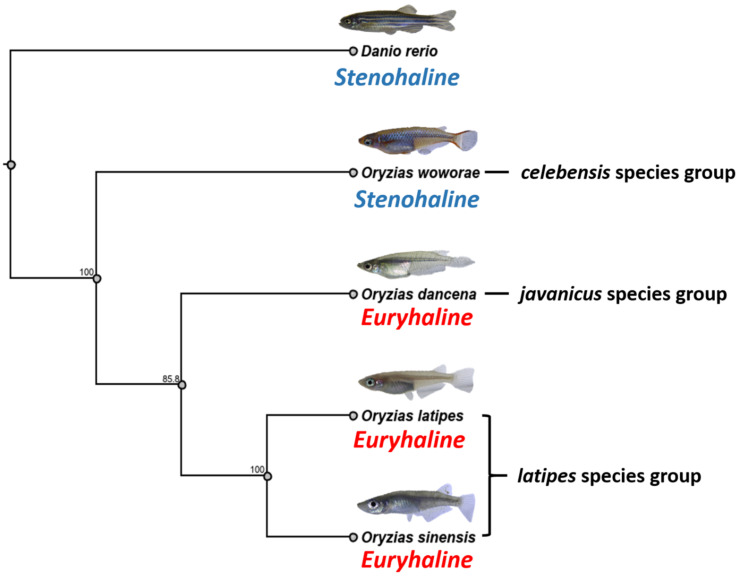
Phylogenetic relationships among four *Oryzias* species and *Danio rerio* inferred from the datasets of NADH dehydrogenase subunit 2 gene (*nd2*). The phylogenetic relationships were analyzed by the maximum likelihood method. The numbers beside the branches indicate the bootstrap values. Based on the prior publication by Murata et al. (2019), *O. latipes* and *O. sinensis* genetically belong to the *latipes* species group while *O. dancena* and *O. woworae* belong to *javanicus* and *celebensis* species groups, respectively.

**Table 1 ijms-22-05686-t001:** Comparison of neurotransmitters, antioxidant activity, and oxidative stress contents in each medaka species brain tissue measured using enzyme-linked immunosorbent assay (ELISA). The data are expressed as the mean with SD. Kruskal–Wallis test continued with uncorrected Dunn’s test was used to analyze the data. Different SSD letters (a, b, c) represent significant statistical differences (*p* < 0.05; *n* = 10 for all groups; * SSD = statistically significant difference).

Biomarkers	*O. dancena*		*O. latipes*		*O. woworae*		*O. sinensis*		Unit
	Concentration	SSD * Letter	Concentration	SSD Letter	Concentration	SSD Letter	Concentration	SSD Letter	
5-HT (Serotonin)	36.21 ± 11.25	a	25.95 ± 5.634	b	40.59 ± 7.444	a	27.16 ± 7.525	b	ng/total protein (mg)
CAT (Catalase)	15.83 ± 5.214	a, c	10.16 ± 2.194	b	14.35 ± 2.980	a	10.77 ± 2.829	b, c	ng/total protein (mg)
NE (Norepinephrine)	1.512 ± 0.6271	a	0.8307 ± 0.2756	b	1.653 ± 0.5085	a	1.011 ± 0.4499	b	ng/total protein (mg)
DA (Dopamine)	22.86 ± 9.282	a	12.84 ± 3.738	b	24.93 ± 5.888	a	14.45 ± 4.942	b	pg/total protein (mg)
Cortisol	228.1 ± 69.08	a	135.5 ± 44.60	b	283.3 ± 68.17	a	146.9 ± 68.29	b	pg/total protein (mg)
AChE (Acetylcholinesterase)	29.82 ± 6.940	a, c	17.09 ± 9.654	b	35.92 ± 7.873	a	23.44 ± 15.40	b, c	U/total protein (mg)
ROS (Reactive Oxygen Species)	133.5 ± 44.48	a, c	80.81 ± 31.65	b	199 ± 69.21	a	89.78 ± 44.00	b, c	IU/total protein (mg)

## Data Availability

The data presented in this study are available in Appendix A and further inquiries can be directed to the corresponding authors.

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
