# Peer review of "Interspecies Behavioral Variability of Medaka Fish Assessed by Comparative Phenomics"

_ijms, 2021, doi:10.3390/ijms22115686_

Round 1
Reviewer 1 Report
This article describes several behaviours in four madaka fish and compare the results between the four species and to zebra fish results. It seems to be original in describing these behaviours in new medaka species. The authors further attempted to relate the phenotypic description to molecular genotype by using Principal Component Analysis and hierarchical clustering analysis, as well as suggesting a phylogenetic analysis based on the sequence of the NADH dehydrogenase subunit 2 gene (ND2) gene.
General comments:
Species names should be in italics font & written in full when first introduced (e.g Line 153).
It will also help to have the common name next to the scientific name when first introduced and specifically as you exchanged between both from time to time (e.g. used both D. rerio or Zebra fish in different areas of the article).
Abstract:
Lines 28-31 - Hard to follow the long sentence between “interestingly” to “not displayed.
Not all genomic results are mentioned.
Introduction:
Line 52 – not clear what you meant by the word “edges” – re-word.
Line 69 – some of the behaviours listed here later described in methods, but some of these terms are not clear to average reader. Consider bring these explanations forward, pecifically the term “Shoaling” needs explanation here.
Lines 72-75 – if the Medaka’s CNS system is simple, this contrast its usefulness as a model to the human complex CNS system.
Line 79 – “this fish”: did you mean Medaka or Zebra fish?
Lines 100-101 “nowadays”: delete the first “a”
Reference missing throughout the introduction: Lines 47-50, 62-63, 71, 84-94, 97, 106-108, 112-114, 114-117
Lines 133-135 – better fits to the discussion/conclusion section?
Lines 135-136 – better fits to the methods?
Results
While p-values are listed in the supplementary tables, it is hard to follow which comparisons done and which of these were significant. E.g.,
Lines 157-159 – Were the “higher average speed” significant in O. dansena?
Titles are confusing, not clear which “comparison” made, between the four Medaka inter-species or each of the four to Zebra fish only?
Lines 247-250 – see above introduction remark on “Shoaling”
Lines 255-256 – to discussion section
Lines 303-314 – Most of this discussion fits better to discussion section
Table 1 – List full name for the abbreviations in the left column
- Not clear what ‘a’ ‘b’ and ‘c’ represents – what the different between these?
- To whom the comparison was? Is it to Zebra fish?
2.8 PCA clustering, not very clear why you decided on these two clusters? Can see that O. dancena and O. Latipes are clustering, but the rest can be interpreting as different clusters. The second half of this section can go to discussion and might be over-interpretation...
Lines 359-362 – move to discussion.
Figures
When you using “a” “b” etc.’ to represent statistical differences, it is not clear what each of these means? (is it <0.05 vs <0.001?)
Figure 1 - Labelling the different colours is missing (i.e. which colour represents which fish?)
What is “AB”?
Hard to understand which comparisons were made here – only to Zebra fish, or also between the different Medaka species. Also hard to reflect these by looking on the panels. Perhaps label one panel to explain your rational.
Figure 2 – the yellow lines are not seen in some of the fishes? (e.g. panel B O. dancana; panel D D. rerio. This is also true for some of the following figures/panels (please check and correct)
Methods Line 297/Figure 5 legend – not clear why the statistic test was different in G and M panels?
Methods/Figures – why the number of individuals compared from O. Sinensis is lower than the rest? (if it due to starting with 30 fish, but losing them, it still need to be mentioned in discussion/limitations).
Figure 7 – the last sentence does not describe your results, not belong here.
Methods –
4.2 Some repetition between results descriptions and methods when presenting the behaviour traits.
Line 633 – No need to describe the computer hardware (unless this is a super-computer?). & Do not use the “@” symbol if stays.
Lines 668-670, why some results as median, but this one as CI.
Line 675 – which “behavioural endpoints” are based on the prior study?
Lines 680-681 – Delete from “After data..., respectively”
Lines 700-701 – over intrepatation
Lines 708-709 – what about differences of molecular/genotyping aspects?
Figure A1 – part of the study design is with your results. Change the format of this figure, if at all necessary?
Tables A1 to A3 – suggest to use bold font for significant results (will easy to follow these)
Table A3 – Is this table supposed to include data of the Zebra fish as well?
Table A4 – section 6 – which is the light or dark treatment?
References
Ref 4 – is this a website? If yes, need to cite its url.
Some of the references only have one page number – check this (e.g. ref 10, 15, 17, 25, 72, 90).
Ref 23 – delete “pp.”
Ref 88 – delete “JoVE” and brackets (leave full journal name).
Ref 94 – Are the “W” needed to be included in “W566-W570”?
Author Response
Review Report Form 1
Comments and Suggestions for Authors
This article describes several behaviours in four madaka fish and compare the results between the four species and to zebra fish results. It seems to be original in describing these behaviours in new medaka species. The authors further attempted to relate the phenotypic description to molecular genotype by using Principal Component Analysis and hierarchical clustering analysis, as well as suggesting a phylogenetic analysis based on the sequence of the NADH dehydrogenase subunit 2 gene (ND2) gene.
General comments:
Species names should be in italics font & written in full when first introduced (e.g Line 153).
Thank you for the correction. In the revised version, all species names are written in full and in italics when first introduced in the abstract and manuscript. Danio rerio is first introduced in line 21 in Abstract and line 51 in Introduction. Meanwhile, medaka fish’s entire full name is first mentioned in lines 25-26 in Abstract and in line 44 (Oryzias latipes), 117 (Oryzias woworae), 158 (Oryzias dancena), and 165 (Oryzias sinensis).
It will also help to have the common name next to the scientific name when first introduced and specifically as you exchanged between both from time to time (e.g. used both D. rerio or Zebra fish in different areas of the article).
The authors appreciated the suggestion. Therefore, the scientific name of species has been added next to the common name, if it is available when first introduced (e.g. “zebrafish (Danio rerio)” on line 51).
Abstract:
Lines 28-31 - Hard to follow the long sentence between “interestingly” to “not displayed.
Thank you for pointing out this matter. The authors agree that the sentence was too long and it was quite hard to follow, thus, the sentence has been rewritten to several sentences to avoid a long sentence.
Not all genomic results are mentioned.
The authors understood the reviewer’s point. Here, the genomic result, which is the genomic-based clustering, was conducted to do a comparison between this clustering and phenomic-based clustering. Therefore, this genomic result has been mentioned in Abstract, specifically in lines 32-35, on discussing the comparison results of the two clusterings. However, additional information regarding the gene used in the genomic-based clustering, which was the ND2 gene, has been added to Abstract.
Introduction:
Line 52 – not clear what you meant by the word “edges” – re-word.
Thank you for the suggestion. It is true that the word “edges” was not suitable to be used in the sentence. Therefore, the word has been changed to “advantages” to avoid confusion and misinterpretation.
Line 69 – some of the behaviours listed here later described in methods, but some of these terms are not clear to average reader. Consider bring these explanations forward, pecifically the term “Shoaling” needs explanation here.
The authors appreciated the suggestion. It is true that some terms, specifically the term “Shoaling”, need an explanation. Therefore, the description of the term has been added to the sentence on lines 74-75. However, since most of the listed behaviors are commonly tested in other animal models, especially rodents, and not all of the listed behaviors are tested in the current study, the authors believe that the description of each behavior listed in the sentence is unnecessary to be explained in this part.
Lines 72-75 – if the Medaka’s CNS system is simple, this contrast its usefulness as a model to the human complex CNS system.
Thank you for pointing out this matter. Actually, the “simple” term mentioned here has a very broad meaning that depends on the reader’s perspective. Besides of medaka fish’s CNS similarity to the human’s, it is definitely more simpler than the human one. However, compared to other commonly used animals, it is more complex and similar to the human CNS than the other animal CNS. Therefore, the authors admitted the usage of this sentence might be confusing for the readers. Thus, to avoid confusion, the sentence was rephrased.
Line 79 – “this fish”: did you mean Medaka or Zebra fish?
The authors appreciated the question. The phrase “this fish” is referred to medaka, not zebrafish. Thus, to avoid confusion and misinterpretation, the words have been changed to “medaka”.
Lines 100-101 “nowadays”: delete the first “a”
Thank you for the input. However, the authors had checked the correct spelling for the word in several credible sources, including the Oxford dictionary, and found that “nowadays” is the correct form of the word. Nevertheless, if this is not addressing the reviewer’s question yet, the authors are opened to another input or explanation.
Reference missing throughout the introduction: Lines 47-50, 62-63, 71, 84-94, 97, 106-108, 112-114, 114-117
The authors understood the reviewer’s point, thus, the authors had checked the missing references, especially on the mentioned lines. However, if there were still some sentences that seem to have no reference, it means that the authors cited more information from those references than other references and put the citation number in the following sentence later.
Lines 133-135 – better fits to the discussion/conclusion section?
Thank you for the suggestion. The authors strongly agreed with the reviewer regarding this matter. Therefore, the sentence in Line 133-135 had moved to the conclusion section since it is more appropriate than before.
Lines 135-136 – better fits to the methods?
The authors appreciated the suggestion. However, since the experimental design in Figure A1 is describing the current study’s outline in general, the authors believe that it is more suitable to be mentioned in the current place, which is in the introduction part, rather than the methods part, considering the specificity of each section discussed in the methods part.
Results
While p-values are listed in the supplementary tables, it is hard to follow which comparisons done and which of these were significant. E.g.,
Thank you for pointing out this matter. Actually, all of the statistic comparisons done in the current study are comparing the data of each group with the data of every other group. Meanwhile, the statistical significance of these comparisons in each figure is shown by letters. Different letters (a, b, c, d) represent a significant difference with a p-value less than 0.05. However, since in the previous version there were no letters to show the statistical significances in Figure 1, the letters were added to make the figure easier to follow as the reviewer suggested.
Lines 157-159 – Were the “higher average speed” significant in O. dansena?
The authors appreciated the question. The answer to the question is yes. From the novel tank test results, O. dancena was found to exhibit higher locomotor activity than O. latipes and O. sinensis. As mentioned in the manuscript, this phenomenon is indicated by statistically higher levels of average speed (p<0.0092 & p<0.0001) and rapid movement ratio (p<0.0001 & p=0.0007) possessed by O. dancena compared to the other fishes.
Titles are confusing, not clear which “comparison” made, between the four Medaka inter-species or each of the four to Zebra fish only?
Thank you for raising this question. The authors strongly agreed to the reviewer’s comment. The previous figure’ captions were indeed confusing since the comparisons made were not clearly described. Therefore, in the current version, the captions of Figure 1-5 and Table 1 were revised to avoid misleading and confusion.
Lines 247-250 – see above introduction remark on “Shoaling”
The authors thanked the reviewer for this constructive comment. As mentioned above, a brief introduction of shoaling behavior was added to the introduction part as the reviewer suggested.
Lines 255-256 – to discussion section
Thank you for the suggestion. The authors feel that there are some differences in terms of line number of the manuscript between the author’s and the reviewer’s since in the author’s version, line 255-256 is the last sentence of Figure 3’s caption. Nevertheless, if the intended sentence is in section 2.5, the authors believe that all of the sentences are more suitable to be placed in the results section since all of the sentences are referring to Figure 4. However, if it is not what the reviewer meant, the authors are welcomed for another input or explanation.
Lines 303-314 – Most of this discussion fits better to discussion section
The authors appreciated the suggestion. As mentioned in the point above, the differences in terms of line number of the manuscript between the author’s and the reviewer’s caused the authors harder to understand the reviewer’s intended sentence. However, the authors had also checked the whole paragraph on section 2.6 and feel that all of the sentences are more suitable to be placed in the current section since all of the sentences are referring to Figure 5. Nevertheless, if it is not what the reviewer meant, the authors are welcomed for another input or explanation.
Table 1 – List full name for the abbreviations in the left column
- Not clear what ‘a’ ‘b’ and ‘c’ represents – what the different between these?
- To whom the comparison was? Is it to Zebra fish?
Thank you for the correction. The authors strongly agreed to the reviewer’s suggestion. Thus, the full name of each abbreviation in Table 1 was added in the left column. Next, the letters in the “SSD letter” column (a, b, c) represent significant differences of each group after compared with the data of every other group. In this result, each medaka fish species was compared to every other medaka fish species, thus, there is no data of zebrafish in this table. All of this information was added to Table 1’s caption.
2.8 PCA clustering, not very clear why you decided on these two clusters? Can see that O. dancena and O. Latipes are clustering, but the rest can be interpreting as different clusters. The second half of this section can go to discussion and might be over-interpretation...
The authors appreciated the reviewer for pointing out this matter. As mentioned in the caption of Figure 6, the two major clusters in PCA were based on the hierarchical clustering analysis results in Figure 6B, which also showed two major clusters. These circles were drawn to make the readers easier to interpret the heatmap clustering data and compare them to the PCA results. Furthermore, the authors also agreed with the reviewer’s suggestion regarding the change of several sentences from section 2.8. Therefore, the second half of the section (was moved to the discussion part. In addition, the authors believe that it is not over-interpretation since the authors only wrote necessary information that supports the results.
Lines 359-362 – move to discussion.
Thank you for the suggestion. As mentioned in the point above, the line number differences in the manuscript caused the authors unsure about the reviewer’s intended sentence. However, if the intended sentence is “Interestingly, this genetic-based… and D. rerio” in section 2.9, the authors had moved the sentence to the discussion part as the reviewer suggested. Nevertheless, if it is not what the reviewer meant, the authors are welcomed for another input or explanation.
Figures
When you using “a” “b” etc.’ to represent statistical differences, it is not clear what each of these means? (is it <0.05 vs <0.001?)
The authors appreciated the question. The different letters in Figures 1-5 and Table 1 represent a significant statistical difference with a p-value less than 0.05. The information regarding the letters and their meaning in terms of statistical significance is mentioned in each caption of Figure 1-5 and Table 1.
Figure 1 - Labelling the different colours is missing (i.e. which colour represents which fish?)
Thank you for the correction. There was a mistake regarding the colors label in Figure 1, thus, the label was added in Figure 1, specifically in Figure 1I (black: D. rerio, red: O. dancena, blue: O. latipes, green: O. woworae, and purple: O. sinensis).
What is “AB”?
The authors appreciated the question. AB mentioned in the test is referring to one of the commonly used zebrafish strains. It is not an abbreviation so there is no full name of AB.
Hard to understand which comparisons were made here – only to Zebra fish, or also between the different Medaka species. Also hard to reflect these by looking on the panels. Perhaps label one panel to explain your rational.
Thank you for pointing out this matter. As mentioned above, the comparisons made here were data of each group with the data of every other group. Therefore, in Figure 1-5, each tested fish species (four medaka fishes and zebrafish) was compared to each other. To avoid confusion, Figure 1’s caption was changed and letters were added to each figure’s panels to help the reader reflect or interpret the data and observe the significant statistical differences in each panel.
Figure 2 – the yellow lines are not seen in some of the fishes? (e.g. panel B O. dancana; panel D D. rerio. This is also true for some of the following figures/panels (please check and correct)
Thank you for the input. It is true that some of the median and interquartile lines were not clearly shown in some figures. The authors had tried to superimpose the lines on top of the individual data, but unfortunately, the graphing software that the authors used cannot do the function. Therefore, to overcome this problem, the size of individual data was reduced and the thickness of the lines was increased, thus, the authors hope that the lines are easier to be seen in the current version of the manuscript.
Methods Line 297/Figure 5 legend – not clear why the statistic test was different in G and M panels?
The authors appreciated the question. However, the statistic test conducted for the data in G and M panels is the same statistic test conducted for the data in other panels in Figure 5, which is Kruskal-Wallis test continued with uncorrected Dunn’s test. This information is mentioned in Figure 5’s caption.
Methods/Figures – why the number of individuals compared from O. Sinensis is lower than the rest? (if it due to starting with 30 fish, but losing them, it still need to be mentioned in discussion/limitations).
Thank you for raising this question. Actually, the authors used a slightly few number of O. sinensis because, during the time, the availability of this fish was very low in the author’s region. Therefore, the authors could not afford O. sinensis in the same sample size number as the other fishes. However, based on the previous studies, even though it is slightly fewer, this sample size number is still appropriate enough to be used in the current experiment. As the reviewer suggested, this crucial information was added to the Materials part, specifically in section 4.2, since the authors believe that this information is more suitable to be placed in that section.
Figure 7 – the last sentence does not describe your results, not belong here.
The authors strongly agreed with the reviewer. It is true that the last sentence in Figure 7 did not belong there. Therefore, the authors decided to remove this sentence as it is already mentioned in the discussion part.
Methods –
4.2 Some repetition between results descriptions and methods when presenting the behaviour traits.
Thank you for pointing out this matter. Indeed, the authors admitted that there were some repetitions between result descriptions and methods when presenting the behavior traits. Therefore, to overcome this problem, some essential information regarding the behavioral tests in the materials section was moved to the Results section while the redundant information in the materials section was removed.
Line 633 – No need to describe the computer hardware (unless this is a super-computer?). & Do not use the “@” symbol if stays.
The authors strongly agreed with the reviewer. The desktop computer used in the current study was not a super-computer and it does not need to be described in the methods section. Therefore, the computer hardware was removed from the text as the reviewer suggested.
Lines 668-670, why some results as median, but this one as CI.
Thank you for the question. It is true that while most of the graphs in Figure 5 are expressed as median with interquartile range, Figure 5G and M are presented as mean with a 95% CI. This decision was taken since the median of some groups is 0, therefore, if the data from that group are expressed as median with interquartile range, the bar will be difficult to be seen because it is in the same line with X-axis. This essential information was added to the figure’s caption.
Line 675 – which “behavioural endpoints” are based on the prior study?
The authors appreciated the question. As mentioned in section 4.5, all of the behavioral endpoints calculated in the present study are based on the prior study. It means that all of the behavioral endpoints measured in the novel tank, mirror biting, predator avoidance, conspecific social interaction, shoaling, and circadian rhythm locomotor activity assays are based on the previous publications mentioned in the manuscript.
Lines 680-681 – Delete from “After data..., respectively”
Thank you for the suggestion. The authors strongly agreed to the reviewer’s suggestion, therefore, the unnecessary sentence in the last line of section 4.5 was removed from the manuscript.
Lines 700-701 – over intrepatation
The authors understood and agreed to the reviewer’s argument. Thus, the sentence “suggesting that… might be required” is removed from that section since it was over interpretation.
Lines 708-709 – what about differences of molecular/genotyping aspects?
Thank you for addressing this question. Actually, the molecular field is also one of the main interests in future studies to have the relationship between the genetic, behavior, and biochemistry of other medaka fish strains. Therefore, the last sentence in the conclusions section was revised as the reviewer suggested.
Figure A1 – part of the study design is with your results. Change the format of this figure, if at all necessary?
The authors appreciated the suggestion. It is true that there is a part of the current study result in the study design figure. Therefore, Figure A1 was revised to avoid redundancy as the reviewer suggested.
Tables A1 to A3 – suggest to use bold font for significant results (will easy to follow these)
Thank you for the constructive suggestion. The authors strongly agreed with the reviewer. Thus, to make the readers easier to understand the tables, all of the P-values that less than 0.05 are written in bold.
Table A3 – Is this table supposed to include data of the Zebra fish as well?
The authors appreciated the question. However, as mentioned above, the biomarkers analysis was conducted only in medaka fish since it is the main concern of the current study, which also applies the 3R principle. Therefore, Table A3 is not supposed to include data of zebrafish.
Table A4 – section 6 – which is the light or dark treatment?
Thank you for pointing out this matter. In the circadian rhythm locomotor activity test, the behavioral endpoint code starts with 6-1 is referring to the behavior endpoint in the day cycle while the behavioral endpoint code starts with 6-2 is referring to the behavior endpoint in the night cycle. This crucial information was added to Table A4 and thanks to the reviewer.
References
Ref 4 – is this a website? If yes, need to cite its url.
The authors appreciated the question. Actually, to the best of the authors’ knowledge, this reference is not a website. Even though it is available online, it is commonly known to be published as a book.
Some of the references only have one page number – check this (e.g. ref 10, 15, 17, 25, 72, 90).
Thank you for the detailed correction. The authors had checked all of the references in the manuscript and corrected some references that only have one-page number as the reviewer suggested.
Ref 23 – delete “pp.”
The authors appreciated the correction. The word “pp” was removed from the reference as the reviewer’s suggestion.
Ref 88 – delete “JoVE” and brackets (leave full journal name).
Thank you for the correction. As the reviewer suggested, “JoVE” and brackets were removed from the references, leaving the full journal name.
Ref 94 – Are the “W” needed to be included in “W566-W570”?
The authors appreciated the question. The authors had checked the reference again and found out that the page numbering in that journal is started with “W”. Therefore, the authors believe that it is more appropriate and suitable to follow the page numbering of that journal.
Reviewer 2 Report
ijms-1227283 “Interspecies Behavioral Variability of Medaka Fish Assessed by Comparative Phenomics”.
GENERAL COMMENT:
The work entitled “Interspecies Behavioral Variability of Medaka Fish Assessed by Comparative Phenomics” is a good work; the subject is original and of current interest.
This study compared the behavioural performance and biomarker expression in the brain between four medaka fishes, which were Oryzias latipes, O. dancena, O. woworae, and O. sinensis, in order to provide more information regarding its behaviour and to demonstrate the behavioural differences between several species of medaka.
The results of the study showed that each medaka species explicitly exhibited different behaviours to each other, which might be related to the different basal levels of several biomarkers.
The subject of the study is interesting and topical.
Central argument is supported by evidence and analysis.
The methodology described by the author is very accurate.
This work is a good work and in my opinion it does not need to be modified.
DETAILED COMMENT:
- Title
-The title is adequate.
- Abstract
The abstract is well structured and the objective of the study is clearly described.
- Introduction
The introduction section is very exhaustive.
- Results
This section is detailed and well written
- Discussion
The discussion section is adequately discussed and exhaustive.
- Tables and figures
Tables and Figures are clear and understandable.
- References
The references are adequate.

Author Response
Review Report Form 2
Comments and Suggestions for Authors
ijms-1227283 “Interspecies Behavioral Variability of Medaka Fish Assessed by Comparative Phenomics”.
GENERAL COMMENT:
The work entitled “Interspecies Behavioral Variability of Medaka Fish Assessed by Comparative Phenomics” is a good work; the subject is original and of current interest.
This study compared the behavioural performance and biomarker expression in the brain between four medaka fishes, which were Oryzias latipes, O. dancena, O. woworae, and O. sinensis, in order to provide more information regarding its behaviour and to demonstrate the behavioural differences between several species of medaka.
The results of the study showed that each medaka species explicitly exhibited different behaviours to each other, which might be related to the different basal levels of several biomarkers.
The subject of the study is interesting and topical.
Central argument is supported by evidence and analysis.
The methodology described by the author is very accurate.
This work is a good work and in my opinion it does not need to be modified.
DETAILED COMMENT:
- Title -The title is adequate.
- Abstract The abstract is well structured and the objective of the study is clearly described.
- Introduction The introduction section is very exhaustive.
- Results This section is detailed and well written
- Discussion The discussion section is adequately discussed and exhaustive.
- Tables and figures Tables and Figures are clear and understandable.
- References The references are adequate.
We appreciate reviewer’s comments and thank you for your precious time on reviewing our paper.
Reviewer 3 Report
The manuscript entitled "Interspecies behavioral variability of medaka fish assessed by comparative phenomics" describes the comparative investigation of behavior and neurochemical markers in four medaka species and zebrafish.
The authors found that, although there were clear individual differences between the medaka species, there was also a clustering of the medaka species compared to zebrafish.
I find that this paper provides relevant information for the IJMS readership on the behavioral and neurochemical characteristics of an increasingly popular group of model fish species. The findings of this study would be of interest to the toxicology community as well as for biomedical and ecological research. However, the manuscript requires some significant revision and rewriting before it is ready for publication.
Below are some general comments followed by specific comments organized by section:
General
- Page numbering- requires formatting
- Grammatical review is required
- Citations and referencing- many references, especially in the introduction section, are not relevant to the information they are cited for. In addition, some of the information does not have appropriate references. Below I list most of the referencing issues that I found
Abstract
- Line 19- rewrite ‘fewer studies were done in medaka compared to zebrafish’
- Line 19- rewrite ‘especially with regard to its behavior’
- Line 20- ‘behaviour’ is the UK spelling. This manuscript is written following the US English style, therefore is should be spelled ‘behavior’
- Line 23- remove ‘which were’
- Line 24- remove ‘from the results’
Introduction
- Lines 42-44- this information should be supported by additional references (in addition to reference [2])
- Line 48- rewrite ‘high fecundity’
- Lines 50-51- references [1, 5-7] are not relevant to the statement regarding cancer research
- Line 52- consider replacing the word ‘edge’ with ‘advantages’
- Line 55- ‘adaptation of photoperiod, higher salinity’. These are presented as advantages of medaka over zebrafish. It is unclear why they are considered advantages.
- Line 55- rewrite ‘and wide temperature range’
- Lines 53-56- reference [6] is more appropriate than [7] for the information in these lines
- Lines 61-65- this sentence is unclear and needs rephrasing
- Line 65- ‘These advantages’- it is not clear which advantages the authors are referring to
- Line 71- ‘their high resemblance to humans’- this statement requires a citation
- Lines 72-73- reference [10] does not support the statement in this sentence
- Lines 75-77- reference [15] is more relevant
- Lines 84-98- please provide citations for this information
- Lines 112-114- reference [19[ should be provided here
- Line 120- please consider starting a new paragraph here
Results
- General comment- species names should be italicised
- Lines 140-141- rewrite ‘Four behavioral endpoints were used’
- Line 142- please explain what is meant by ‘swimming’ as a behavioral endpoint
- Line 151- replace ‘Danio rerio’ with ‘zebrafish’
- Line 159- rewrite ‘different movement types’
- Figure 1- explain the color scheme in the graphs
- Line 188- rewrite relative interaction time’
- Lines 230-232- please explain in more detail what each measured parameter means
- Lines 248-249- rewrite ‘Shoaling, an innate behavior for several fish to swim together, was observed in each medaka fish’
- Line 271- reference [28] is not relevant to the statement
- Figure 5A- indicate clearly that the time intervals on the x-axis are in hours
- Line 309- ‘these neurotransmitters’- explain which specific neurotransmitters are indicated
- Line 309- AChE- explain the abbreviation
- Line 316- CAT, ROS- explain the abbreviations
- Table 1- ‘Biomarkers’ column- explain abbreviations
- Line 335- rewrite ‘formed’ as ‘formation’
- Figure 6 caption, line 348- rewrite ‘different medaka species and zebrafish’
- Figure 6 caption, line3 349-350- rewrite ‘another fish species (Danio rerio)’ as ‘zebrafish’
Discussion
- Lines 403-404- the authors describe the mirror assay as ‘a social behavior model’, however it is more widely viewed as a test of aggression. Please clarify.
- Lines 419-420- Is this a hypothesis proposed by the authors? If yes, please clarify this in the text
- Lines 422-423- Please clarify what is meant by ‘supported by the variation in catalase’s basal levels’. Was the variation in catalase levels directly correlated with ROS levels/fish behavior?
- Lines 436-437- this sentence is not grammatically correct
- Lines 451-452- was this observed in the study results? Please clarify
- Lines 475-476- was this statement supported by the study results? Please clarify
- Lines 490-491- the sentence is not grammatically correct
- Line 492- please clarify the abbreviation ‘LD cycles’
- Lines 490-507- it seems that this section is not relevant to the study and it is unclear to me how the information in this section supports or explains the results of the study. Please clarify or remove this section
- Lines 519-521- the relevance of this sentence is unclear
- Line 531- rewrite ‘these phenomena might be related to the different’
- Paragraph in lines 536-576- the discussion in this paragraph seems out of context and it is unclear how this is related to the results of the study, i.e. the behavioral and neurochemical results
- Paragraph in lines 577-585- here again it is unclear how the calculate coefficient supports/relates to the results of the study, and how it may be used in future studies (designing/ planning experiments?)
Methods
- Section 4.2. Animal Ethics and Behavioral Tests-
- please provide brief descriptions of the behavioral test procedures
- what was the timeline of the behavioral testing? Was each test done on a separate day or were there multiple tests done in each testing day?
- Line 625- rewrite ‘Generally, a fish displays’
- Lines 632-633- computer specifications are not necessary

Author Response
Review Report Form 3
Comments and Suggestions for Authors
The manuscript entitled "Interspecies behavioral variability of medaka fish assessed by comparative phenomics" describes the comparative investigation of behavior and neurochemical markers in four medaka species and zebrafish.
The authors found that, although there were clear individual differences between the medaka species, there was also a clustering of the medaka species compared to zebrafish.
I find that this paper provides relevant information for the IJMS readership on the behavioral and neurochemical characteristics of an increasingly popular group of model fish species. The findings of this study would be of interest to the toxicology community as well as for biomedical and ecological research. However, the manuscript requires some significant revision and rewriting before it is ready for publication.
Below are some general comments followed by specific comments organized by section:
General
- Page numbering- requires formatting
The authors appreciated the input. The page numbering was re-formatted as the reviewer suggested.
- Grammatical review is required
Thank you for the reminder. The authors had tried their best to do a grammatical review. The authors hoped that the grammatical error is significantly reduced in this version of the manuscript.
- Citations and referencing- many references, especially in the introduction section, are not relevant to the information they are cited for. In addition, some of the information does not have appropriate references. Below I list most of the referencing issues that I found
The authors thanked the reviewer for the detailed correction. As mentioned below, all of the references were corrected according to the reviewer’s suggestion.
Abstract
- Line 19- rewrite ‘fewer studies were done in medaka compared to zebrafish’
Thank you for the suggestion. The sentence was rewritten to “fewer studies were done in medaka compared to zebrafish” as the reviewer suggested.
- Line 19- rewrite ‘especially with regard to its behavior’
The authors appreciated the suggestion. The sentence was rewritten to “especially with regard to its behavior” according to the reviewer’s suggestion.
- Line 20- ‘behaviour’ is the UK spelling. This manuscript is written following the US English style, therefore is should be spelled ‘behavior’
Thank you for pointing out this matter. It is true that there were several mistakes regarding the spelling of some words. Therefore, as the reviewer suggested, all of the words’ spellings in the manuscript were changed to follow the US English style.
- Line 23- remove ‘which were’
The authors appreciated the suggestion. As the reviewer suggested, the phrase ‘which were’ was removed.
- Line 24- remove ‘from the results’
Thank you for the suggestion. As the reviewer suggested, the phrase ‘from the results’ was removed.
Introduction
- Lines 42-44- this information should be supported by additional references (in addition to reference [2])
The authors thanked the reviewer for the suggestion. As the reviewer suggested, several additional references were added to the sentence regarding the use of medaka as a model organism.
- Line 48- rewrite ‘high fecundity’
Thank you for the suggestion. The phrase was rewritten to ‘high fecundity’ as the reviewer suggested.
- Lines 50-51- references [1, 5-7] are not relevant to the statement regarding cancer research
The authors understood the reviewer’s point. Actually, the authors believe that two references are relevant to the statement regarding cancer research as supported by some figures below from some parts of these references. However, as the reviewer mentioned, the other two references are indeed not relevant to the statement, thus, these two references were moved.
- Line 52- consider replacing the word ‘edge’ with ‘advantages’
Thank you for the suggestion. The word ‘edge’ was replaced with ‘advantages’ as the reviewer suggested.
- Line 55- ‘adaptation of photoperiod, higher salinity’. These are presented as advantages of medaka over zebrafish. It is unclear why they are considered advantages.
The authors appreciated the reviewer’s question. These traits of medaka considered as advantages because these traits allow researchers to be able to design broader experiments with more variations and options. In addition, these traits also make medaka easier to be kept and handled.
- Line 55- rewrite ‘and wide temperature range’
Thank you for the suggestion. The phrase was rewritten to ‘wide temperature range’ according to the reviewer’s suggestion.
- Lines 53-56- reference [6] is more appropriate than [7] for the information in these lines
The authors agreed with the reviewer. Therefore, the reference [6] was moved to the previous sentence, and reference [7] was removed from the sentence.
- Lines 61-65- this sentence is unclear and needs rephrasing
Thank you for the suggestion. It is true that the sentence is unclear, thus, the sentence was rephrased according to the reviewer’s suggestion.
- Line 65- ‘These advantages’- it is not clear which advantages the authors are referring to
The authors appreciated the reviewer’s comment. Indeed, the reference for the phrase ‘These advantages’ was not clear enough. Therefore, the whole sentence was revised and merged with other previous sentences.
- Line 71- ‘their high resemblance to humans’- this statement requires a citation
Thank you for the reminder. Some references were added to the statement as the citations regarding the medaka high genetic resemblance to humans.
- Lines 72-73- reference [10] does not support the statement in this sentence
The authors appreciated the reviewer’s input. It is true that the reference does not support the statement in the sentence. Therefore, the reference was removed and the sentence was revised.
- Lines 75-77- reference [15] is more relevant
Thank you for the suggestion. The reference mentioned by the reviewer was added to the end of the sentence ‘Additionally, medaka also… those of amniotes’.
- Lines 84-98- please provide citations for this information
The authors thanked the reviewer for the reminder. Several citations for the information were added to the manuscript.
- Lines 112-114- reference [19[ should be provided here
Thank you for the correction. The mentioned reference was added to that particular line.
- Line 120- please consider starting a new paragraph here
The authors strongly agreed with the reviewer’s suggestion. Therefore, a new paragraph is added, starting from ‘In the present study…”.
Results
- General comment- species names should be italicized
Thank you for the reminder. The authors had double-checked the species name throughout the manuscript and typed them in italic.
- Lines 140-141- rewrite ‘Four behavioral endpoints were used’
The authors thanked the reviewer for the input. Therefore, the phrase was rewritten according to the reviewer’s suggestion.
- Line 142- please explain what is meant by ‘swimming’ as a behavioral endpoint
Thank you for pointing out this matter. Actually, the word ‘swimming’ meant swimming movement time ratio, however, the phrase ‘movement time ratio’ was put at the end of the sentence. Therefore, to avoid confusion, all of the behavioral endpoint names are written in the full name.
- Line 151- replace ‘Danio rerio’ with ‘zebrafish’
The authors appreciated the suggestion. The words ‘Danio rerio’ was replaced with ‘zebrafish’ according to the reviewer’s suggestion.
- Line 159- rewrite ‘different movement types’
Thank you for the input. The phrase was rewritten to ‘different movement types’ as the reviewer suggested.
- Figure 1- explain the color scheme in the graphs
The authors thanked the reviewer for the detailed correction. In this current version, the information regarding the color scheme in Figure 1 was added to the figure.
- Line 188- rewrite relative interaction time’
Thank you for the input. The words were rephrased to ‘relative interaction time’ as the reviewer suggested.
- Lines 230-232- please explain in more detail what each measured parameter means
The authors appreciated the suggestion. However, since the detail of each measured behavior parameter is already detailed described in the supplementary Table A4, the authors believe that there will be redundancies if the details are added to the paragraph.
- Lines 248-249- rewrite ‘Shoaling, an innate behavior for several fish to swim together, was observed in each medaka fish’
Thank you for the suggestion. The sentence was rewritten as the reviewer’s suggestion.
- Line 271- reference [28] is not relevant to the statement
The authors appreciated the input. However, the authors believe that the reference is relevant to the statement. This argument is supported by a paragraph from the reference shown below. “Several recent investigations in mammals have made it virtually certain that the daily rhythm of gross locomotor activity, often used as a convenient assay of circadian rhythmicity, is governed by at least two coupled oscillators” in our citated article.
- Figure 5A- indicate clearly that the time intervals on the x-axis are in hours
Thank you for the input. The authors admitted that the time intervals on the x-axis were not clearly indicated. Therefore, additional information was added to indicate that the time intervals are in hours.
- Line 309- ‘these neurotransmitters’- explain which specific neurotransmitters are indicated
The authors appreciated the suggestion. Therefore, the intended neurotransmitters mentioned in the sentence were explained.
- Line 309- AChE- explain the abbreviation
Thank you for the suggestion. The abbreviation of AChE was added to the manuscript as the reviewer suggested.
- Line 316- CAT, ROS- explain the abbreviations
The authors appreciated the input. The abbreviations of CAT and ROS were added to the manuscript according to the reviewer’s suggestion.
- Table 1- ‘Biomarkers’ column- explain abbreviations
Thank you for the suggestion. All abbreviations in Table 1, specifically in ‘Biomarkers’ column were explained, as the reviewer suggested.
- Line 335- rewrite ‘formed’ as ‘formation’
The authors thanked the reviewer for the detailed correction. The word ‘formed’ was rewritten as ‘formation’ according to the reviewer’s suggestion.
- Figure 6 caption, line 348- rewrite ‘different medaka species and zebrafish’
Thank you for the input. The sentences were rewritten to ‘different medaka species and zebrafish’ as the reviewer suggested.
- Figure 6 caption, line3 349-350- rewrite ‘another fish species (Danio rerio)’ as ‘zebrafish’
The authors appreciated the input. Therefore, the words were rewritten to ‘zebrafish’ according to the reviewer’s suggestion.
Discussion
- Lines 403-404- the authors describe the mirror assay as ‘a social behavior model’, however it is more widely viewed as a test of aggression. Please clarify.
Thank you for the correction. As the reviewer mentioned, mirror assay is more widely viewed as a test of aggression. However, aggression itself is also one of the social behaviors, which became the reason why the authors wrote the sentence in the first place. Nevertheless, to avoid confusion, the sentence was rewritten according to the reviewer suggested.
- Lines 419-420- Is this a hypothesis proposed by the authors? If yes, please clarify this in the text
The authors appreciated the question. The sentence in the mentioned line is indeed a hypothesis proposed by the authors. Therefore, the sentence was rewritten to clarify this matter according to the reviewer’s suggestion.
- Lines 422-423- Please clarify what is meant by ‘supported by the variation in catalase’s basal levels’. Was the variation in catalase levels directly correlated with ROS levels/fish behavior?
Thank you for pointing out this matter. Actually, the sentence was meant to state that the variation in catalase levels might directly be correlated with fish behaviors, not ROS levels. Therefore, the sentence was rewritten to avoid misinterpretation as the reviewer suggested.
- Lines 436-437- this sentence is not grammatically correct
The authors appreciated the correction. The sentence was rewritten to avoid grammatical error.
- Lines 451-452- was this observed in the study results? Please clarify
Thank you for the question. Actually, the sentence is based on the specific basal serotonin content of each medaka fish observed in the study results. Therefore, the sentence was rewritten accordingly.
- Lines 475-476- was this statement supported by the study results? Please clarify
The authors appreciated the question. Actually, the statement regarding the dopamine and norepinephrine levels is based on the study results, therefore, the sentence was rewritten to avoid confusion.
- Lines 490-491- the sentence is not grammatically correct
Thank you for the correction. The sentence was rewritten to avoid grammatical error.
- Line 492- please clarify the abbreviation ‘LD cycles’
The authors appreciated the suggestion. The abbreviation of ‘LD cycles’, which is light-dark (LD) cycles, was added to the manuscript.
- Lines 490-507- it seems that this section is not relevant to the study and it is unclear to me how the information in this section supports or explains the results of the study. Please clarify or remove this section
Thank you for the input. The authors agreed with the reviewer, thus, several unnecessary information regarding the circadian rhythm of medaka were removed. However, there is still some important information that remains since this information supports the results of the study. This information including the typical diurnal circadian locomotor activity rhythms displayed by O. latipes that might be related to the feeding schedule since the rhythm is entrainable by feeding time.
- Lines 519-521- the relevance of this sentence is unclear
The authors understood the reviewer’s point of view. However, from the authors’ perspective, the sentence is relevant to support the results. The information described in the sentence is to show that each fish with its own specific behavior possesses a different level of biomarkers in brain tissue, which helps to support the current hypothesis.
- Line 531- rewrite ‘these phenomena might be related to the different’
Thank you for the input. The sentence was rewritten as the reviewer suggested.
- Paragraph in lines 536-576- the discussion in this paragraph seems out of context and it is unclear how this is related to the results of the study, i.e. the behavioral and neurochemical results
The authors appreciated the reviewer’s comment. However, the authors believe that the discussion is necessary to be mentioned in the manuscript since in the present study, not only behavioral and neurochemical results are displayed, but also the phenomic and genomic results. Therefore, this paragraph is not out of context since it discusses the phenomic and genomic results, especially their differences and similarities to each other, obtained in the current study.
- Paragraph in lines 577-585- here again it is unclear how the calculate coefficient supports/relates to the results of the study, and how it may be used in future studies (designing/ planning experiments?)
Thank you for the question. Here, the authors were not only intended to display the behavioral differences between tested medaka fishes, but the authors also wanted to show the data reproducibility of each medaka behavior. The data reproducibility is important for future studies since it helps the researchers to decide the sample size used, which is closely related to animal usage ethics. In addition, the coefficient of variation also supports the validity of the behavior results and the statistical powers in the current experiment.
Methods
- Section 4.2. Animal Ethics and Behavioral Tests-
- please provide brief descriptions of the behavioral test procedures
The authors appreciated the reviewer’s suggestion. The brief descriptions of the behavioral test procedures were added to the manuscript, specifically in section 4.2., as the reviewer suggested.
- what was the timeline of the behavioral testing? Was each test done on a separate day or were there multiple tests done in each testing day?
Thank you for raising out these questions. Here, the first five behavioral tests were divided into three sessions, which were conducted in a single week. The first session contained novel tank and shoaling tests while the second session consisted of mirror biting and social interaction tests. Meanwhile, the predator avoidance test belonged to the last session of these tests. Furthermore, the circadian rhythm locomotion test was conducted the following week. The background for this test order as described in the manuscript.
- Line 625- rewrite ‘Generally, a fish displays’
Thank you for the input. However, this sentence was removed from the manuscript since it was suggested by the other reviewer.
- Lines 632-633- computer specifications are not necessary
The authors strongly agreed with the reviewer. The authors agreed that the specifications of the desktop computer used in the current study did not need to be described in the methods section. Therefore, the computer hardware was removed from the text as the reviewer suggested.